# T-DOpE probes reveal sensitivity of hippocampal oscillations to cannabinoids in behaving mice

Jongwoon Kim[1,4], Hengji Huang[1,4], Earl T. Gilbert[2,4], Kaiser C. Arndt [2], Daniel Fine English[2] ✉ & Xiaoting Jia [1,2,3] ✉

Understanding the neural basis of behavior requires monitoring and manipulating combinations of physiological elements and their interactions in behaving animals. We developed a thermal tapering process enabling fabrication of low-cost, flexible probes combining ultrafine features: dense electrodes, optical waveguides, and microfluidic channels. Furthermore, we developed a semi-automated backend connection allowing scalable assembly. We demonstrate T-DOpE (Tapered Drug delivery, Optical stimulation, and Electrophysiology) probes achieve in single neuron-scale devices (1) high-fidelity electrophysiological recording (2) focal drug delivery and (3) optical stimulation. The device tip can be miniaturized (as small as 50 μm) to minimize tissue damage while the ~20 times larger backend allows for industrial-scale connectorization. T-DOpE probes implanted in mouse hippocampus revealed canonical neuronal activity at the level of local field potentials (LFP) and neural spiking. Taking advantage of the triple-functionality of these probes, we monitored LFP while manipulating cannabinoid receptors (CB1R; microfluidic agonist delivery) and CA1 neuronal activity (optogenetics). Focal infusion of CB1R agonist downregulated theta and sharp wave-ripple oscillations (SPW-Rs). Furthermore, we found that CB1R activation reduces sharp wave-ripples by impairing the innate SPW-R-generating ability of the CA1 circuit.

Recent advancements in neural interface technology have significantly improved our understanding of the nervous system[1–10], presenting new opportunities to explore systems-level neurophysiology, the treatment of neurological disorders, and the development of brain-machine interfaces. However, the complexity of the brain's inner workings demands increased precision in the monitoring and manipulation of neural activity in order to decipher and understand its connectivity and dynamics[1,11–14]. Optogenetics has been developed as a powerful tool to modulate and/or monitor neural activities with a high spatiotemporal resolution and cell-type specificity[15,16]. It has led to the widespread adoption of bi-directional devices, which are able to monitor electrical activity and apply optical stimulation[4,17–20]. However, other biological factors such as neurochemistry are intertwined with electrical activity, necessitating simultaneous opto-electro-pharmacological investigations. One such example, is the interaction of exogenous cannabinoids with neural circuitry. With the recent rise in popularity of cannabinoids due to widespread legalization in the United States, this pharmacological-electrophysiological interaction needs to be further investigated. Combining drug infusion with current opto-electric devices has remained challenging. Therefore, developing single devices that can interact with the brain of behaving mammals across such multiple modalities is a critical goal for the field.

[1]The Bradley Department of Electrical and Computer Engineering, Virginia Tech, Blacksburg, VA, USA. [2]School of Neuroscience, Virginia Tech, Blacksburg, VA, USA. [3]Department of Materials Science and Engineering, Virginia Tech, Blacksburg, VA, USA. [4]These authors contributed equally: Jongwoon Kim, Hengji Huang, Earl T. Gilbert. ✉e-mail: neurodan83@vt.edu; xjia@vt.edu

A few technologies have been developed aiming to address this challenge[21,22], however, the fabrication of these probes is mostly based on cleanroom microfabrication techniques which are time-consuming and expensive. Here, we develop a probe technology by merging two cutting edge fabrication approaches, i.e., thermal fiber drawing and multi-material tapering. This scalable probe technology opens opportunities for broad distribution of flexible opto-electro-pharmacological probes in the neuroscience community. In addition, a wide selection of new materials, geometric arrangement, and functionalities can be embedded in these tapered fiber probes.

More specifically, the thermal fiber drawing method has emerged as a promising technique for producing scalable multimodal fiber devices at a low cost since 2015[23–29]. Such fiber devices are fabricated via a method commonly used in industry to produce optical fibers. The macroscale, multi-material preform is heated until softened, and pulled into hundreds of meters of fiber that can be as thin as a human hair. The fast and simple fabrication process utilizing affordable machinery and soft material results in a cheap, sturdy and biocompatible device. However, an inherent challenge of fiber technology lies in the uniform diameter across the length: there is a tradeoff between a minimized sensing tip for biocompatibility and a maximized backend tip for easy connection. This has limited the fiber's practicality in neuroscience applications. To overcome this challenge, we developed a thermal tapering process and a semi-automated connection method, which enable us to fabricate microprobes with high structural and functional complexities and scalability (Fig. 1, left). Using this approach, the connectorization is no longer a limiting factor in our probe design. It is noteworthy that over the past decades, the tapering of silica has enabled significant advancement in the neuroscience field such as glass micropipettes used in patch clamp recording. More recently, tapered silica fiber was developed for multisite photometry[30] and single neuron recording[8]. In this work, we demonstrated multi-material tapering to create multiplexed, multifunctional neural probes. This Tapered Drug delivery, Optical Stimulation, and Electrophysiology (T-DOpE) probe allows us to investigate highly complex neural circuitry such as that of the hippocampus in behaving mice performing tasks.

The hippocampus plays a major role in memory, including for learned spatial locations[31,32]. During exploration, hippocampal area CA1 (CA1) pyramidal cell (PYR) activity is organized at the theta timescale (6–11 Hz)[33,34]; during consummatory behaviors and non-rapid eye movement sleep, large depolarization events drive the generation of sharp wave-ripples (SPW-R, 100–250 Hz). Coordinated activity of PYR organized first by theta followed by SPW-R associated replay is thought to support memory consolidation[35]. Further, SPW-Rs are an important electrophysiological marker of learning and memory[36,37] and causal roles for SPW-Rs in driving specific behaviors have been demonstrated. Behavioral performance is improved when SPW-Rs are extended in duration[38]; while disruption/truncation of SPW-Rs in both wake and non-REM sleep states decreases performance[39–42]. In rodents, cannabinoid type-1 receptor (CB1R) activation leads to neuronal populations losing their temporally structed co-activity, thought to cause the disruption of hippocampal synchronous activity including epileptic seizures (high frequency oscillations)[43–46], as well as theta oscillations and SPW-Rs[47,48]. The cannabinoid induced disruption of theta and SPW-Rs is suggested to be a mechanism behind cannabinoid-associated memory impairment in rodents[47,49–52] and humans[53–55]. While a mechanism behind seizure disruption has been identified[56], the relationship between the CB1R and memory-supporting rhythms remains unclear. It is believed cannabinoids act through CB1R to impair memory by changing the spiking activity of neurons, either through acting directly on cells expressing CB1R in CA1, or indirectly by acting on presynaptic partners[52,57–62]. Thus, the precise mechanisms by which CB1R agonists impair hippocampal rhythms have yet to be identified.

Theta and SPW-R oscillations require independent mechanisms of generation[33–35,37,63], and CB1R expression widely varies across cell types and brain areas[51,58–60,64,65], further complicating our understanding of specific cellular/synaptic loci at which cannabinoid signaling occurs. Among CA1 neurons, CB1 receptors are expressed on the axon terminals of CCK+ basket cells, which act to suppress neurotransmitter release[58,59,66], and on pyramidal cell dendrites, decreasing excitability[67]. The theta oscillation in CA1 is strongly supported by inputs from the medial septum[33,63], where CB1Rs are widely expressed including on

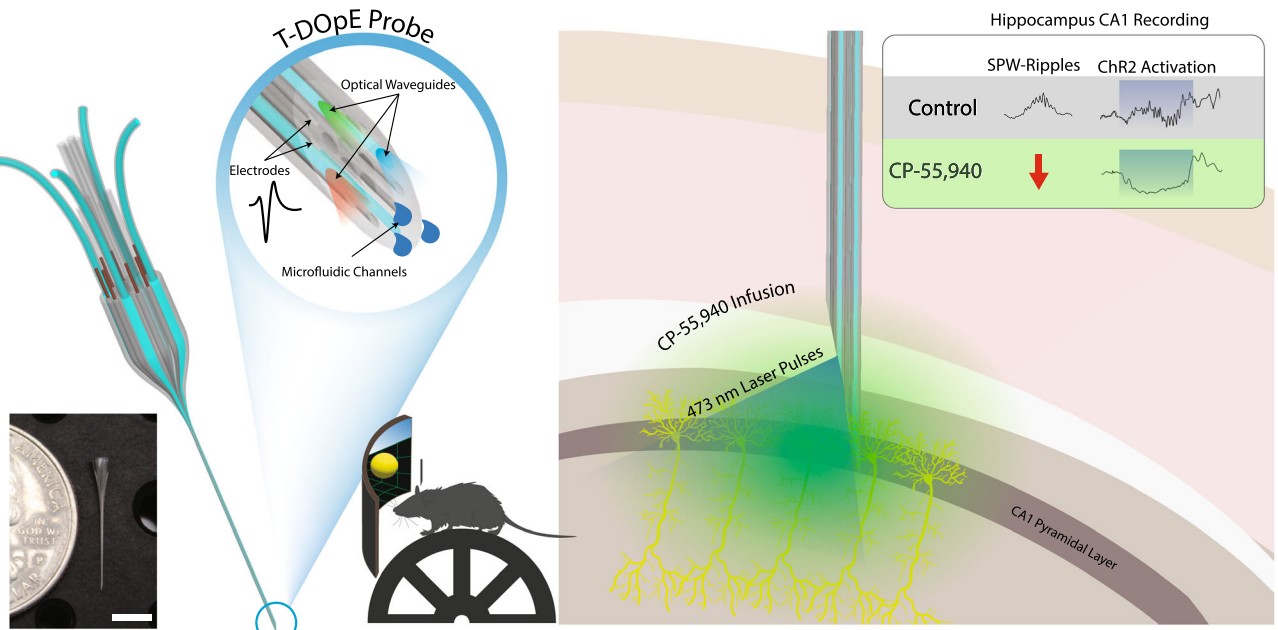

**Fig. 1 | T-DOpE probe and CA1 electrophysiological activity in response to local chemical manipulations under natural and optically stimulated conditions.** T-DOpE probes offer higher complexities at the sensing tip while easing the connection between the backend and the electronics. The probe is implanted into behaving head-fixed mice expressing channel rhodopsin (ChR2) in pyramidal neurons in hippocampal area CA1. Local infusion of synthetic cannabinoid (CB1 Agonist, CP-55,940) in CA1 is sufficient to abolish both spontaneous and optogenetically induced sharp wave-ripples (SPW-R) (Scale bar: 5 mm).

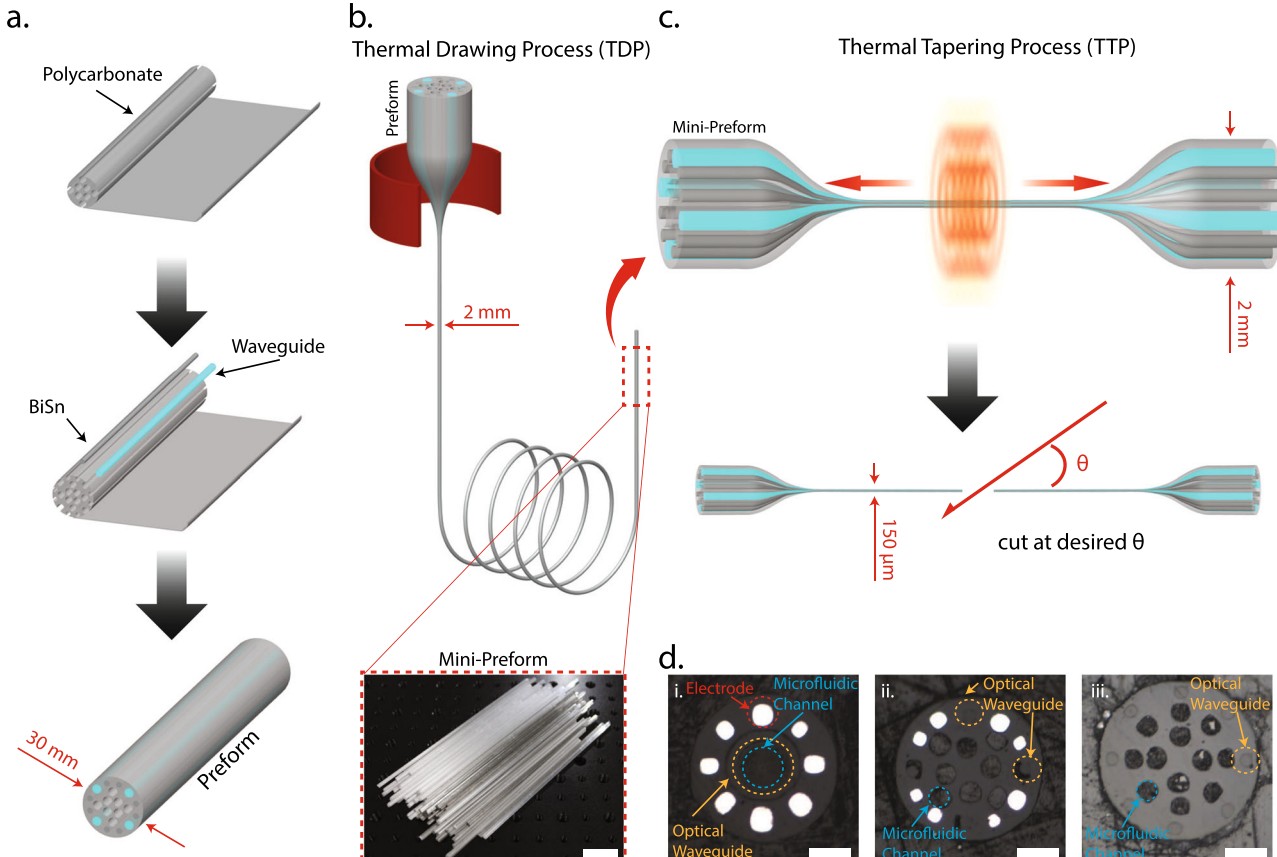

**Fig. 2 | T-DOpE probe fabricated using thermal tapering process. a** Schematic of the preform fabrication. **b** Thermal drawing process pulling 30 mm diameter preform into 2 mm diameter fibers. The fibers are cut into 10 cm long mini-preforms for tapering (Scale bar: 2 cm) **c** Illustration of the thermal tapering process of mini preform. Similar to glass pipette pulling, the mini preform is heated until softened, pulled and cut at a desired angle, resulting in two individual probes. **d** Cross-sectional images of the three various designs. (i) Eight electrodes, one microfluidic channel, and one optical waveguide. (ii) Eight electrodes, eight microfluidic channels, and four optical waveguides. (iii) Twelve microfluidic channels and eight optical waveguides. (Scale bar: 50 μm).

axon terminals projecting to CA1[64]. Systemic infusion of CB1 agonists disrupt the theta oscillation and organization of spike timing[47,48], but because this method activates CB1Rs throughout the body and brain it is unable to identify mechanistic loci. Thus, it is currently unclear if the activation of CB1Rs within the medial septum, within CA1, or on projections from the medial septum are responsible for theta disruption. The SPW-R oscillation is generated by the reciprocal interaction of excitatory pyramidal cells and local inhibitory interneurons in CA1[68–71], initiated by strong excitatory input from Schaffer collaterals originating in CA3[35,37], whose terminals express CB1 receptors[60–62]. SPW-Rs can also be generated experimentally by depolarizing CA1 excitatory neurons[68]. The cellular-synaptic mechanisms by which cannabinoids disrupt SPW-Rs have remained enigmatic in large part due to the challenging nature of monitoring and manipulating neuronal activity with pharmacological interventions in microcircuits of behaving animals, which our probes robustly support. Specifically, our devices are able to deliver the synthetic cannabinoid CP-55,940, a validated tool to study the effect of CB1R activation on neuronal activity[47–49,52], to a local circuit which we monitor with electrophysiology and manipulate further with optogenetics.

Here, we present the Thermal Tapering Process (TTP) which enables high feature density, functional complexity, and semi-automated backend connections. Various designs of the probe were fabricated to demonstrate the capabilities achievable with the tapering method. To test our device performance in vivo, T-DOpE probes were implanted in awake, behaving mice. We demonstrate that T-DOpE probes can reliably record hippocampal circuit electrophysiology in both acute and chronic platforms. We then demonstrated precisely controlled optical and chemical modulation of neural activity using our probes in CA1. We use our device to test the hypothesis that activation of CB1 receptors in CA1 weakens the theta oscillation during exploration. Additionally, we tested if activating CB1 receptors expressed by CA1 neurons is sufficient to disrupt SPW-Rs. We found that intra-CA1 administration of CP-55,940 during head-fixed navigation was sufficient to reduce theta oscillations and abolish spontaneous SPW-Rs. Interestingly, we discovered that the local infusion of CP-55,940 in CA1 of a mouse expressing channel rhodopsin (ChR2) in CA1 pyramidal cells inhibits optogenetically induced SPW-Rs (Fig. 1, right). For the first time, we used a single device with unique multimodal capabilities, to examine the effects of CA1 cannabinoid receptor activation on local oscillatory activity. We found that regardless of CA3 inputs, CB1R agonism disrupts SPW-R generation, emphasizing the importance of cannabinoid signaling for local circuit computation.

## Results

### T-DOpE probe fabricated using thermal tapering process

Multifunctional tapered probes with various designs were fabricated via the Thermal Drawing Process (TDP) followed by a Thermal Tapering Process (TTP). Figure 2a shows the fabrication steps for a preform. A Polycarbonate (PC) rod was machined to create hollow channels (microfluidic channels in the finished device), and spaces to inlay waveguides and electrodes. Waveguides with a PC core (refractive index $n = 1.59$) and Poly(methyl 2-methylpropenoate) (PMMA; refractive index $n = 1.49$) cladding and bismuth tin (BiSn) alloy electrodes

were then inserted into their respective positions. The preform was then wrapped in PC film and consolidated in a vacuum furnace. Through the TDP, the finalized preform was heated and drawn down to a 2 mm diameter fiber (Fig. 2b). The fiber diameter was closely monitored via a laser micrometer and controlled by adjusting the pulling speed and temperature. The fiber was cut into 10 cm long mini-preforms (~one hundred per TDP draw) for the subsequent thermal tapering process. During TTP, mini-preforms are heated, softened, and pulled to form a tapered structure (Fig. 2c). The process is adapted from the fabrication method of heating glass pipettes and tapered silica optical fibers[72]. Here, instead of using a single material (silica) as in the traditional tapering approaches, we develop a multi-material thermal tapering approach to co-draw multiple functional components inside a polymer matrix. The pulled structure was then cut at a specific angle, resulting in two individual microprobes with 2 mm diameter backends and 150 μm tapered tips for monitoring and manipulating neural activity.

By varying the temperature and the pulling speed, thicker (200 μm in diameter) or thinner tapered fiber (50 μm in diameter) can be achieved, as shown in Supplementary Fig. 1. To meet the impedance requirement for high-quality neural recording, the electrodes were designed to be at least 25 μm in diameter. Drug delivery channels can be as small as 20 μm without collapsing (Supplementary Fig. 2). However, the pressure required to push fluid through a microfluidic channel is proportional to the channel radius[4] and fluid viscosity. For this study in which we delivered vehicle and CP-55,940 which has a high viscosity, we chose drug channels with a diameter of ~40 μm to ensure stable and reliable drug delivery. For this study, probes with three different designs (8/1/1, 8/4/8 and 0/8/12; electrodes/waveguides/microfluidic channels, respectively) were fabricated. The size of the electrodes, microfluidic channels, and optical waveguides are ~25 μm, 30–50 μm, and 30 μm respectively. Their device tip cross sections are shown in Fig. 2d. The BiSn electrodes record extracellular voltage while the optical waveguides control optogenetics, and the microfluidic channels enable focal drug infusion.

## T-DOpE probe connections and characterizations

We developed the fast, semi-automated backend connection method to replace the traditional method which is slow and labor-intensive. Our connection to the T-DOpE probe was executed exclusively on its 2 mm backend (Fig. 3a). Electrical connection was accomplished by heating the probe's backend and inserting insulated copper wires into melted electrodes. The probe was then cooled for the BiSn electrodes to solidify. The microfluidic connection was achieved by inserting a custom drawn thin PC tube into the microfluidic channels. Custom drawn polymer optical fibers were coupled onto the waveguides on the probe's backend. Epoxy was then used to seal and fix the microfluidic and optical connections in place. A fully functional and connectorized probe with 8 electrodes, 4 waveguides and 8 microfluidic channels is shown in Fig. 3b. Probes are fitted with industry standard adapters compatible with electrophysiology equipment, optical modules, and drug delivery pumps. A photograph of a probe connected to a printed circuit board (PCB) is shown in Supplementary Fig. 3.

To demonstrate the flexibility and durability of the probe, we bent the probe at a > 45° by applying axial force pressing the tip on a hard surface as shown in Fig. 3c. A video of a tapered probe bending is also available in Supplementary Video 1. We measured the spectral impedance of the T-DOpE probe before and after bending. As shown in Fig. 3d, bending has no statistical impact on our device's impedance at 1 kHz, with an average impedance of $192 \pm 50$ kΩ before bending and $220 \pm 30$ kΩ after bending. (Paired *t*-test, NS: not significant, $p > 0.05$ (After vs. Before, $p = 0.3717$, $n = 8$)). Using a dynamic mechanical analyzer, we measured the bending stiffness of the three designs and stainless steel wire, included in Supplementary Fig. 4. Note that all designs have a much lower stiffness than the stainless steel wire with

the same diameter. Additionally, the elongation to fracture of polycarbonate is 110%, which means, unlike some polymers that shatter like glass, such as acrylic, polycarbonate can undergo large deformation without breaking or cracking. We also demonstrated our optical waveguide's individual addressability in our device with 8 waveguides and 12 microfluidic channels. The cross-sectional images of the device with red and green lights emitting from the waveguides are shown in Fig. 3e. The transmission spectrum of the optical waveguide is included in the Supplementary Fig. 5. To test our device's ability to deliver multiple drugs independently via separate microfluidic channels, we conducted an experiment with a device of the same design in 0.6% agarose gel as illustrated in Fig. 3f. The device was first inserted in the agarose gel and delivered green food dye (Fig. 3g). The blue and orange dye were infused after insertion into deeper regions of the gel. The diffusion of the dyes clearly indicated the infused regions of the agarose gel. The video of this experiment is available in Supplementary Video 2. These measurements and tests verify the functionality of our probe for extracellular recording, optogenetic manipulation, and drug delivery.

## In vivo electrophysiology recording capabilities of T-DOpE probe

To evaluate the functionalities of the T-DOpE probe in vivo, we collected acute and chronic electrophysiological activity in behaving mice. As shown in Fig. 4a, probe implantation was targeted to the CA1 region of the hippocampus. Figure 4b shows an example of an acute wideband (0.1–8000 Hz) extracellular trace from CA1, which demonstrates a transition from rest, when SPW-Rs occur, to a running state, when theta oscillations emerge. Shaded region **i.** demonstrates multi-unit activity in the raw trace (i.e., bandpass filtered at 0.1–8000 Hz). Shaded regions **ii.** and **iii.** indicate oscillations which are established as identifying electrophysiological landmarks of hippocampus CA1[36,37,73]: SPW-Rs (**ii.**, 100–250 Hz), and theta-nested gamma oscillations (**iii.**, Theta: 6–11 Hz, Gamma: 40–80 Hz)[33,34]. T-DOpE probes can thus acquire single units and local field potential (LFP) activity in behaving animals without disrupting natural physiological process within the local circuit.

We additionally chronically implanted the T-DOpE probes in CA1 ($n = 2$ animals) to demonstrate biocompatibility and ability for long-term stable single unit recording capability. Three example putative neurons were identified from recordings on 8/9 and 9/21. Each cell is color coded to match the auto-correlations and the spike waveforms across days. The average waveforms recorded from each electrode are included in Supplementary Fig. 6. The neurons can be classified by the shape of the waveform and autocorrelation[74] (putative ID, red: pyramidal cell, blue: interneuron, and orange: pyramidal cell). The waveforms from each electrode and the auto- and cross- correlation suggest that the probe recorded from the same group of neurons across 43 days. Additionally, the peak in the cross-correlation from the red and blue cells (Fig. 4c*) shows a monosynaptic connection where the red unit is driving the blue unit to fire action potentials[75,76]. The observation that this monosynaptic connection maintained across 43 days further supports the fact that the probe can reliably record from the same population of neurons over at least one month, a common goal in modern probe designs[3,4,77]. Sorted units and their cluster qualities, such as L-ratio and isolation distance, over all the acute sessions are included in Supplementary Table 1. Though multi-unit activity (MUA) and power in the spike frequency band is observed in all recordings, we used Kilosort and manual curation using Phy to detect sortable units in 13 out of 19 acute sessions ($n = 15$ animals). Detected units ranged from 1–7 units (median = 3) with 57.9% of units identified as putative pyramidal cells, and 42.1% as putative interneurons. Validated spike sorting metrics[74] of Isolation Distance ($\mu = 26.9$, $\sigma = 9.7$), and percentage of spikes that violate the inter-spike interval (<2 ms, $\mu = 0.53\%$, $\sigma = 0.61\%$) are reported. These data demonstrate that our T-DOpE

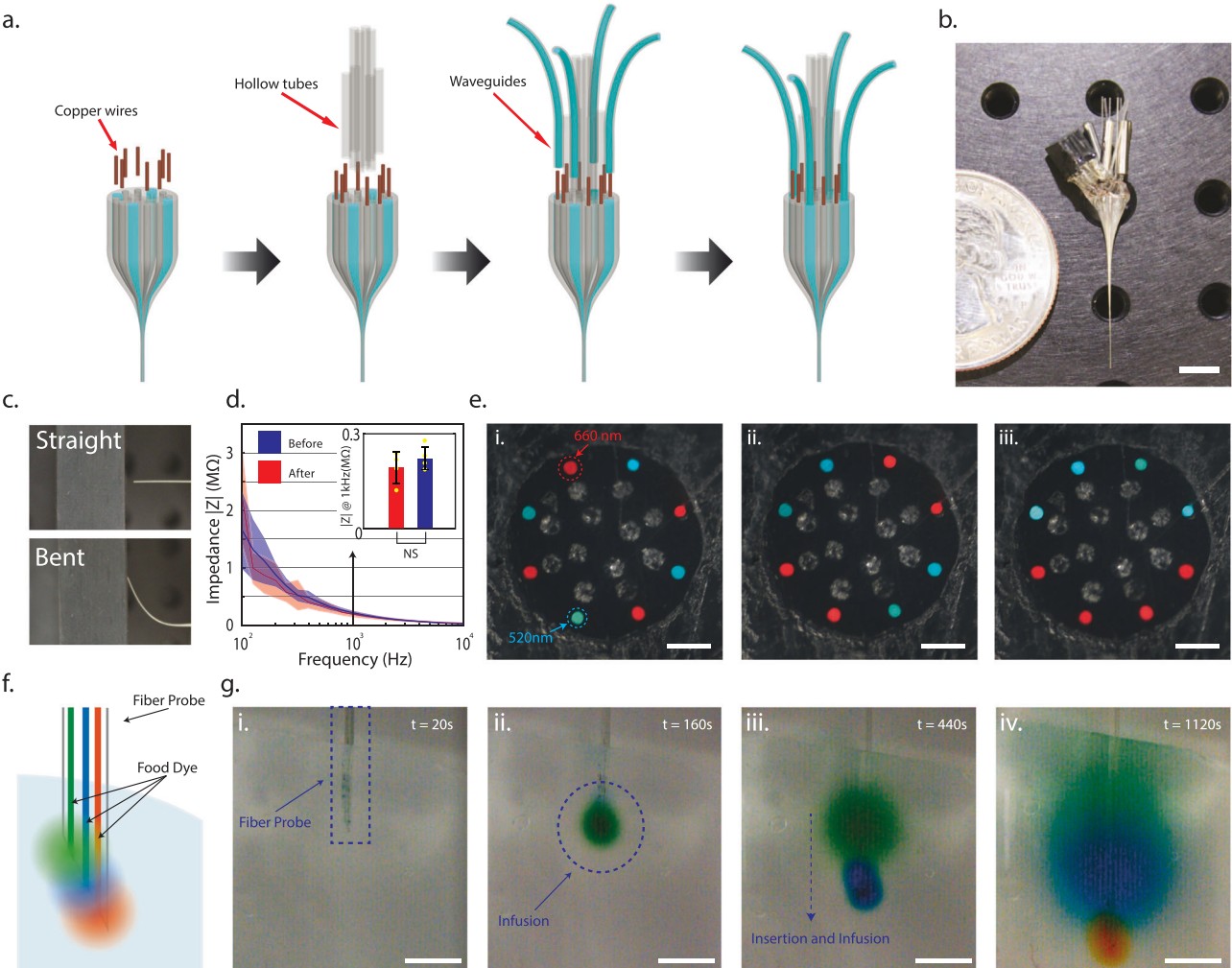

**Fig. 3 | T-DOpE probe connections and characterizations. a** Schematic of the T-DOpE probe backend connection process. **b** Photograph of a fully connected probe with eight electrodes, eight microfluidic channels, and four optical waveguides. (Scale bar: 5 mm) **c** Photograph of a straight probe and a probe bent roughly 45° to demonstrate the flexibility at the sensing end. **d** Impedance measurements of the BiSn electrodes before and after bending. All error bars and shaded colors represent the standard deviation. (Student's two-tailed *t*-test, NS: not significant,

$p > 0.05$ (After vs. Before, $p = 0.3717$, electrode numbers = 8)). Data are presented as mean ± SD. **e** Cross-sectional images to illustrate the individual addressability of the optical waveguides. (Scale bar: 50 μm) **f** Cartoon of green, blue, and orange food dye independently injected via the probe into a brain phantom. **g** Time-lapsed images to demonstrate the drug infusion in a 0.6% agarose gel. The inserted probe demonstrates the infusion of three different dyes at three different heights in the phantom. (Scale bar: 1 mm).

probe has sufficient biocompatibility and stability to perform longitudinal experiments without disrupting local circuitry.

## In vivo modulation capabilities of T-DOpE probe: optogenetic and drug infusion

To validate the optical waveguides in our T-DOpE probe, we recorded from CA1 in mice with ChR2 expression restricted to CA1 pyramidal neurons. In these experiments, the probe is advanced until the four most ventral electrodes are ~100 μm above the center of the CA1 pyramidal layer, with the four most dorsal electrodes remaining in overlying cortex. Figure 5a shows a representative optically evoked neural response from three levels of optical stimulation intensity (i.e., laser power). By regulating optical stimulation power, we were able to modulate individual unit spike rate (Fig. 5b). Note that the lowest optical power does not elicit action potentials, while middle and high level of optical power increase the firing rate. Figure 5c presents examples of local field potential responses to three optical levels. No optically induced responses were observed in cortex, as expected because ChR2 expression was localized to CA1. In CA1, optical stimulation induced large negative deflections in the LFP, likely due to cation

influx through ChR2 and active membrane conductances in the ChR2 expressing neurons, as well as synaptically evoked inward currents in postsynaptic neurons activated by glutamate release from the ChR2+ cells. Optogenetic activation of CA1 PYR induced local high frequency ripple oscillations in CA1[68]. Figure 5c* indicates a SPW-R event endogenously occurring between optical stimulations. These findings demonstrate the probe's ability to monitor and manipulate local circuits while preserving physiological functions of the local circuit.

There is a general problem in drug infusion into the brain in that tissue is displaced, which is especially noticeable when combined with single unit electrophysiology. To validate that the transient silencing of cells is a result of the physical cell displacement[72,78,79] instead of pharmacological influence, we performed a detailed comparison of the dynamic spike recording during the infusion of saline, vehicle, and drug+vehicle. We observed transient absences of spikes after infusion of either saline, vehicle, or drug+vehicle (Fig. 5d, Supplementary Fig. 7). Figure 5e, shows an example session where we infused 200 nL at 1 nLs⁻¹ infusion rate and recorded one putative pyramidal cell (purple) and interneuron (green). After the displaced tissue relaxed to its original position, spikes were once again able to be detected, both

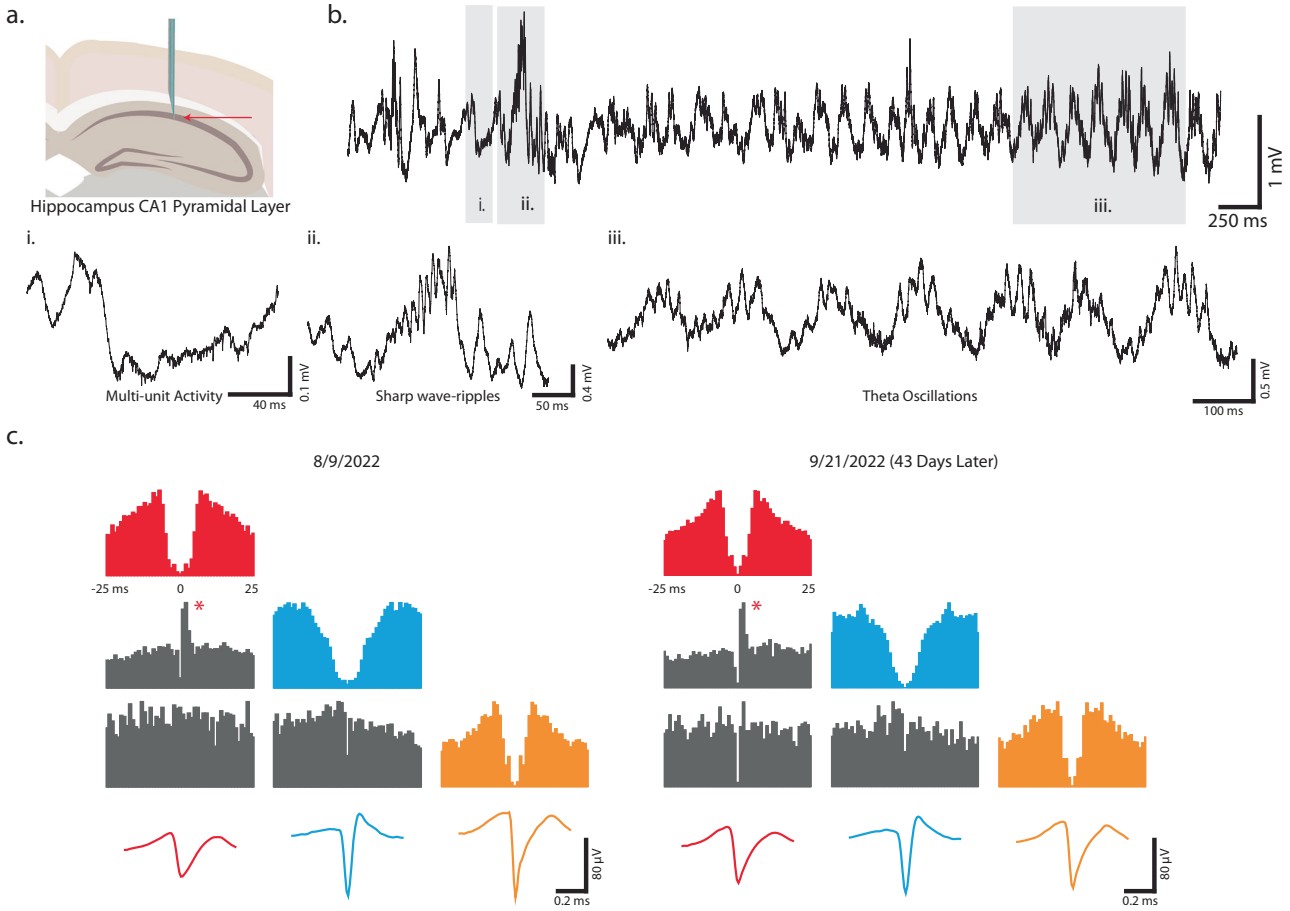

**Fig. 4 | In vivo electrophysiology recording capabilities of T-DOpE probe.** **a** Schematic of the targeted implanted site, hippocampus CA1. **b** Example wideband (0.1–8000 Hz) extracellular traces obtained from CA1. (i) Enlarged trace to display multi-unit activity. (ii) An example trace of a sharp wave-ripple (100–250 Hz) dorsal to the pyramidal layer. (iii) An example of theta-nested gamma oscillations (6–11 Hz; 40–80 Hz). **c** Three example cells recorded on 8/9 and 9/21. Each cell is color coded to match the auto-correlations and the spike waveforms. *Peak in cross-correlation across the red and blue cells suggest their excitatory monosynaptic connection is maintained over 43 days.

at the level of single sorted cells (Fig. 5e, post-infusion) and multi-unit activity (data not shown). Figure 5f, g shows the spike waveforms and autocorrelations of the two cells before and after infusion. The similarities in their waveforms and auto-correlations suggest that these are the same interneuron and pyramidal cell. Importantly, small changes in waveform shape are likely due to the displacement and return of the tissue to the same location with some spatial jitter and thus a change in the effective resistance through the Ohmic brain tissue. However, this jitter is likely in the range of tens of microns, explaining the similarity in baseline and recovery spikes. Of note, this is a major improvement over experimental designs which require drug infusion distant from recording sites, making the locus of effect difficult to interpret.

## CA1 theta power during running is reduced by pharmacological activation of CB1Rs expressed in CA1

To examine the role of CB1R in hippocampal CA1 activity during behavior, we used the T-DOpE probe to focally infuse CB1R agonist, CP-55,940, in CA1 while simultaneously monitoring neural activity (n = 10 awake head-fixed mice). Figure 6a displays an illustration of the experimental setup. Head-fixed mice ran on a vertical wheel driving a 1-D visual virtual reality environment that reliably promotes natural (i.e. spontaneous not due to training) running behavior. Figure 6b shows a representative session of CP-55,940 infusion (16.8 ng; 200 nL; 1 nLs⁻¹) following one-hour recording of baseline (Fig. 6b, c is from one CP-55,940 session). The speed of the mouse is shown above the spectrogram of the session. While mice run, assemblies of place cells

are dynamically synchronized with the theta oscillation (6–11 Hz). The speed data is used to determine the running epochs of the mice, and to demonstrate that mice continue running following infusion. The average normalized speed of each session is shown in Supplementary Table 2. The local infusion of CP-55,940 was sufficient to lower the power of theta oscillations in the LFP (calculated from data restricted to running epochs (Fig. 6b)). The representative power spectrum of baseline (Before infusion; duration: 60 min) and after drug infusion (one hour after infusion; duration: 60 min) are shown in Fig. 6c. Note the peak at 8 Hz decreased by ~20%. Figure 6d shows an example control session with drug vehicle (VEH), the normalized spectrum of the baseline and after VEH infusion. The comparison between baseline and VEH shows no significance while the comparison between the baseline and CP-55,940 showed significant difference (Paired t-test, NS: not significant $p \geq 0.05$, *$p \leq 0.05$ (Baseline vs. VEH, $p = 0.2737$, animal number = 5), (Baseline vs. CP-55,940, $p = 0.0026$, animal number = 5)) (Fig. 6e). Across all sessions, theta power significantly decreased following CP-55,940 infusion compared to vehicle control (n = 5 animals for both VEH, and CP-55,940), with no changes in time spent running or average running speed following infusion (Supplementary Table 2). These findings demonstrate that focal CA1, as opposed to systemic[47], agonism of CB1Rs is sufficient to reduce theta oscillations in CA1. These findings also suggest cannabinoids can impair hippocampal oscillations by acting upon only CB1Rs expressed in CA1 (terminals of inputs from the medial septum and Schaffer Collateral from CA3, or local CA1 neurons), without acting on those in other brain areas.

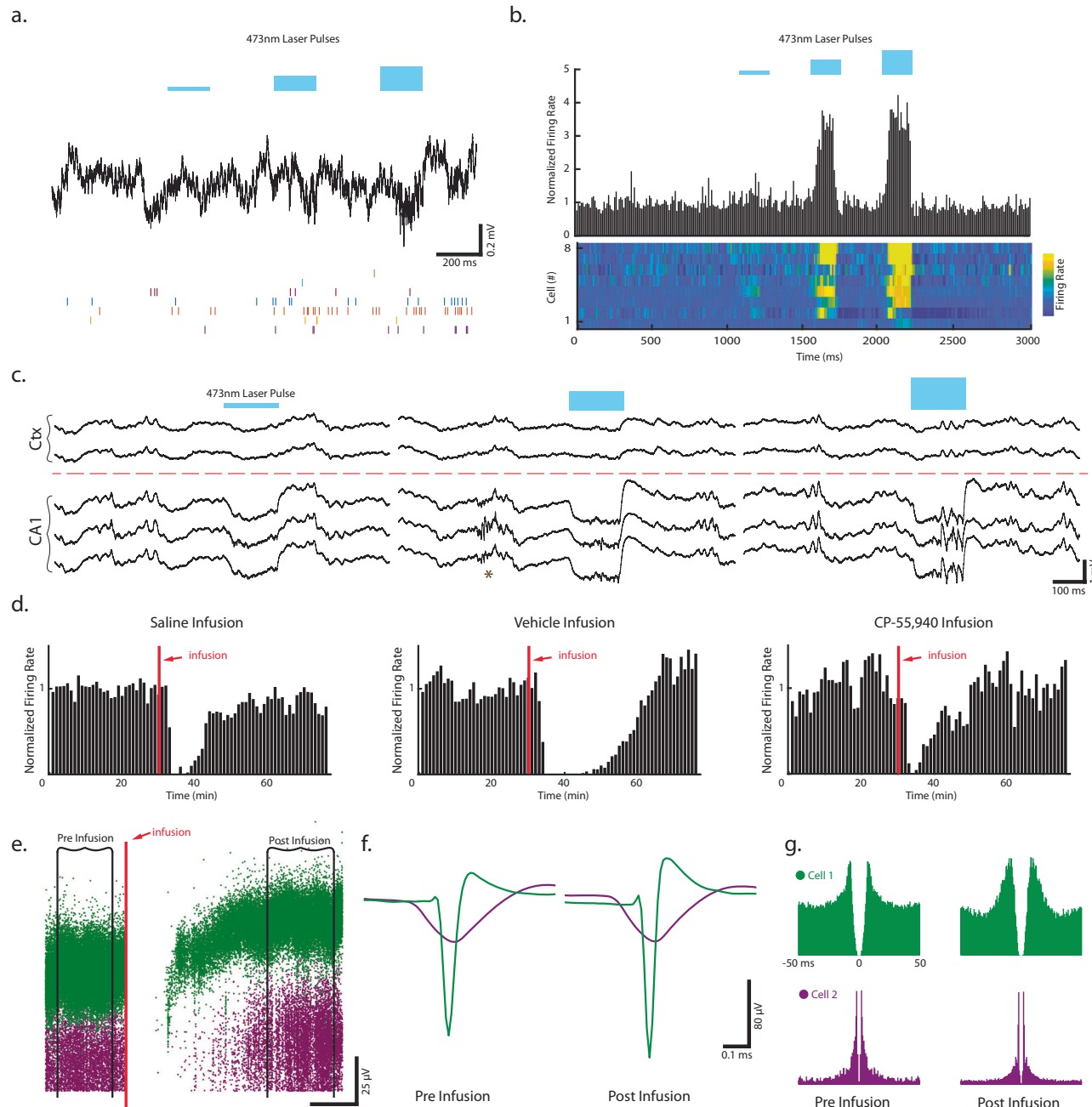

**Fig. 5 | In vivo modulation capabilities of T-DOpE probe: optogenetic and drug infusion. a** Example trace of wideband extracellular response to optical stimulation. **b** The firing rate of sorted cells during optical stimulation and the normalized firing rate across all sessions. **c** Examples of optically evoked responses with three levels of laser power. No response was observed in the cortex due to AAV-ChR2 expression localized to CA1. The amount of optically evoked neuron activity can be increased by varying the light output, allowing us to achieve optically induced SPW-Rs at higher laser powers. * shows spontaneous SPW-Rs in between optical stimulations. **d** Normalized firing rate of sorted cells in saline, vehicle, and drug+vehicle infusion. Note shortly after the infusion, the firing rate of all cells drastically decrease not due to pharmacological influence but to the physical displacement of the cells from the recording sites. **e** Spike amplitude of an interneuron and a pyramidal cell to demonstrate the recovery after infusion (200 nL; 1 nL s⁻¹). Given the device tip contains both the microfluidic channel and recording sites (<20 μm in distance), the cells are pushed away and return after some period of time. **f** Average spike waveforms of the two cells before and after infusion. **g** Autocorrelation of the two cells before and after infusion.

## SPW-R rate is lowered by pharmacological activation of CB1Rs expressed in CA1

Systemic administration of CP-55,940 decreases SPW-R rate and power in behaving animals[47], and that under urethane anesthesia, intrahippocampal infusion of CP-55,940 reduces SPW-R power[47]. Interestingly, we find similar results for local intrahippocampal infusion in CA1 of behaving mice, further implicating local CB1R expression in changes in CA1. With data recorded from the same setup shown in Fig. 6a,

sessions were analyzed in higher frequency bands (100–300 Hz) to study the influence of CP-55,940 on SPW-Rs. In Fig. 7a, the power spectrogram of a CP-55,940 infusion session was computed with the ripple rate plotted on top. The upper panel shows the speed of the mouse. CP-55,940 (16.8 ng; 200 nL; 1 nLs⁻¹) was locally infused at the recording site after 60 min of baseline recordings. After the infusion, the power in 100–300 Hz bins significantly decreased, which indicates a reduction in SPW-Rs. Importantly, ripple rate returned to baseline

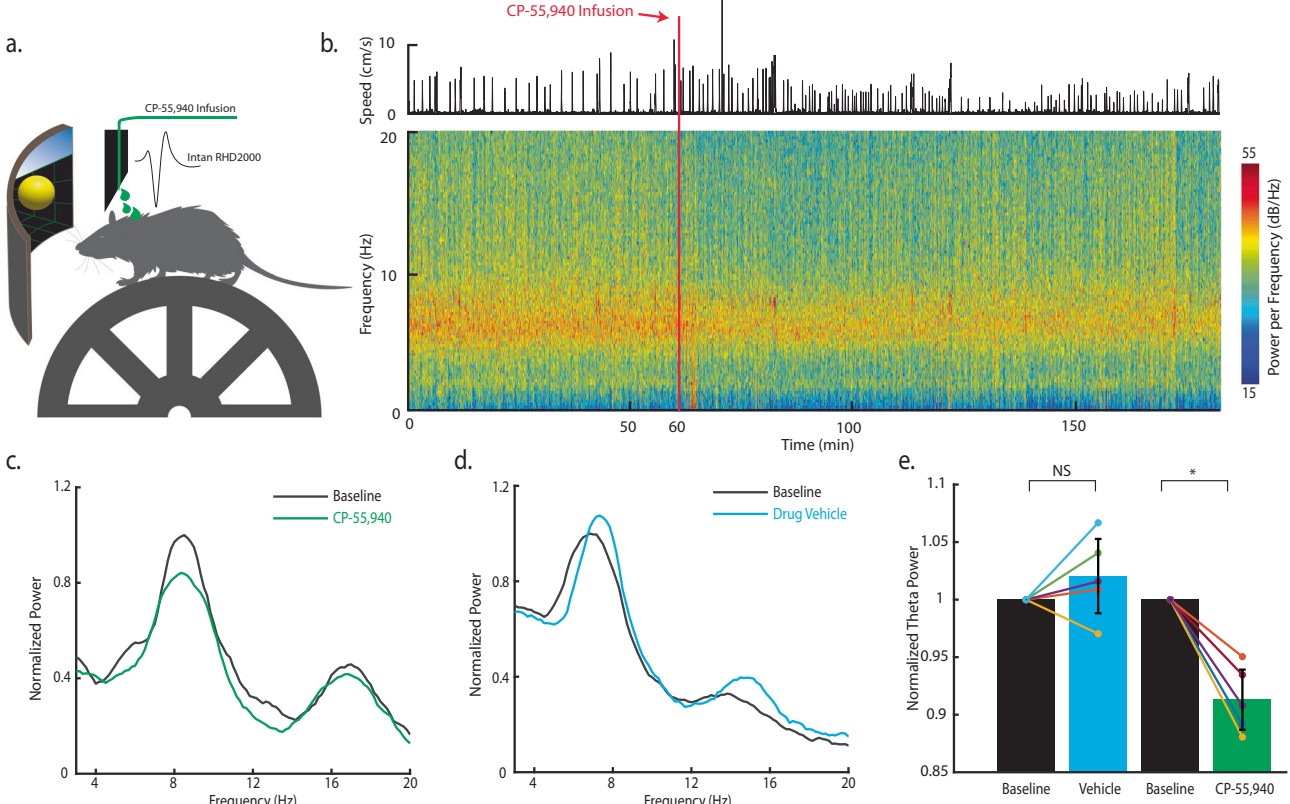

**Fig. 6 | CA1 theta power during running is reduced by pharmacological activation of CB1Rs expressed in CA1. a** Illustration of the experimental setup. A head-fixed, wild-type mouse is mounted on the wheel. A virtual reality environment is presented on the screen to instigate running, and the virtual position is recorded. Local infusion and neural recordings are achieved using the T-DOpE probe. **b** Power spectrogram of CP-55,940 infusion session in lower frequency bins (0–20 Hz). The upper panel shows the speed of the mouse. CP-55,940 (mg kg$^{-1}$; 200 nL; 1 nL s$^{-1}$) is locally infused at the recording sites after 60 min of baseline. **c** Representative normalized power spectrum of the baseline (Before infusion; duration: 60 min) and

after CP-55,940 infusion (one hour after infusion; duration: 60 min) restricted to running time (>2 cm s$^{-1}$ and >1 s running epochs). **d** Representative normalized power spectrum of the baseline (Before infusion; duration: 60 min) and after drug vehicle infusion (one hour after infusion; duration: 60 min) restricted to running time with the same criteria as above. **e** Comparison of the normalized theta power (6–11 Hz) between baseline and infused vehicle or drug (Paired two-sided *t*-test, NS: not significant $p \geq 0.05$, *$p \leq 0.05$ (Baseline vs. Drug Vehicle, $p = 0.2737$, animal number = 5), (Baseline vs. CP-55,940, $p = 0.0026$, animal number = 5)). Data are presented as mean ± SD.

after ~6 hours (Fig. 7b). Spectral analysis of the baseline (Before infusion; duration: 60 min) after CP-55,940 infusion (one hour after infusion; duration: 60 min) were computed. The bump centered at 150 Hz in the baseline spectrum (Fig. 7c) disappeared following CP-55,940 infusion, due to the abolishment of SPW-Rs. The difference of the two spectra was plotted to further visualize the change in the power at each frequency (Fig. 7d). The bump in Fig. 7c is highlighted further by the negative peak at 150 Hz, due to impaired SPW-R generation. The ripple rate (per minute, normalized to ripple rate of the first hour) was calculated to investigate the impact of drug infusion on SPW-R occurrence (Fig. 7e, 10 sessions). We compared the normalized SPW-R count between baseline and infused vehicle or drug in Fig. 7f. (Wilcoxon Signed-rank test, *****$p \leq 0.00001$, NA $p > 0.05$ (Baseline vs. VEH, $p = 0.5857$, animal number = 5), (Baseline vs. CP-55,940, $p = 1.7333e-6$, animal number = 5)). Overall, the injection of CP-55,940, but not VEH, significantly lowered the ripple rate. This demonstrates the importance of local CA1 CB1R signaling in SPW-R dynamics in behaving mice.

### Generation of SPW-Rs by optical stimulation of CA1 PYR is abolished by pharmacological activation of CA1 CB1Rs

SPW-Rs are generated when CA1 receives strong excitatory input from CA3, with the sharp wave resulting from dendritic depolarization of CA1 PYR by CA3, and the fast oscillation due to intra-CA1 interactions between excitatory pyramidal neurons and local GABAergic inhibitory interneurons[37,68,80]. While our results confirm that spontaneous SPW-R

rate decreases following intrahippocampal CP-55,940 infusion, it has remained unclear if this is due to signaling at CB1Rs expressed on synaptic terminals originating from CA3 inputs[51,60,81] and/or by hippocampal interneurons. To address whether changes in CA3 inputs could explain the reduction in SPW-Rs, we substituted the CA3 sharp wave excitation of CA1 PYR with direct ChR2 depolarization[68]. We used the T-DOpE probe to optogenetically depolarize ChR2 + CA1 PYR before and after CP-55,940 infusion. Similar to the setup in Fig. 6a, a head-fixed AAV-CamKII-ChR2 mouse was placed on the wheel with a 473 nm laser connected to the T-DOpE probe to deliver optical pulses into CA1 (Fig. 8a). Following 30 min of baseline, CA1 pyramidal cells were optically stimulated with 150 ms optical pulses at low, medium, and high power over 40 min (Fig. 8b; $n = 400$ stimulations). 10 min after the last optical stimulation, CP-55,940 (16.8 ng; 200 nL; 1 nLs$^{-1}$) was focally infused. Once the tissue recovered from infusion, another set of optical pulses was delivered, using the same optical power levels. The spectrogram of the session was computed with the ripple rate overlayed, as shown in Fig. 8b. Note the increased SPW-R events detected from 40–70 mins due to the optically induced SPW-Rs. After CP-55,940 infusion, the occurrence of SPW-R events drastically decreased and identical low, medium, and high light pulses were not sufficient to optically induce SPW-Rs (Fig. 8c). Optical stimulation activated the network to the same extent before and after, as indicated by the fact that the mean optical LFP responses to identical light pulses were not different (Fig. 8d). Importantly, this demonstrates that failure to

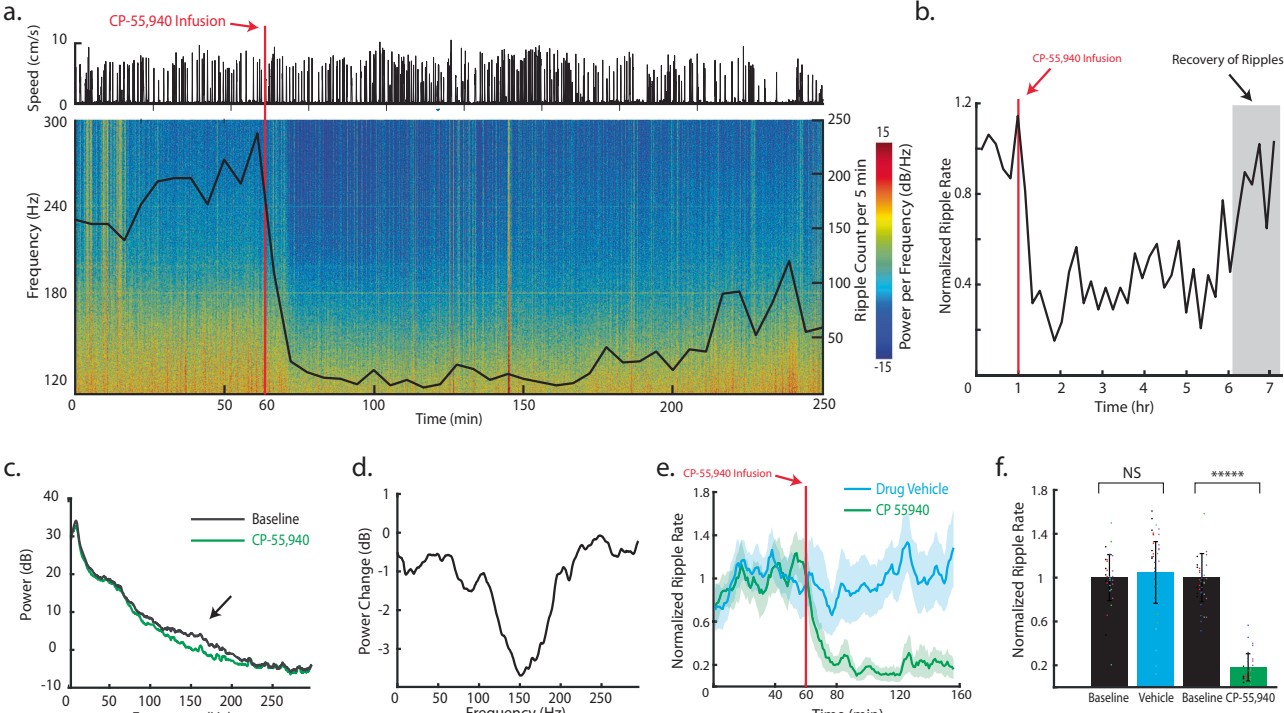

**Fig. 7 | SPW-R rate is lowered by pharmacological activation of CB1Rs expressed in CA1. a** Power spectrogram of CP-55,940 infusion session on higher frequency bins (100–300 Hz) with the same experimental set-up as Fig. 6a. The ripple rate is plotted on top of the spectrogram with respect to the same timescale. The upper panel shows the speed of the mouse. CP-55,940 (mg kg$^{-1}$; 200 nL; 1 nL s$^{-1}$) is locally infused at the recording sites after 60 min of baseline. **b** Ripple count per 10 min over 7-hour recording session. **c** Power spectrum of the baseline (Before infusion; duration: 60 min) and after CP-55,940 infusion (one hour after infusion; duration: 60 min). **d** Spectral difference from the baseline to the recording after CP-55,940 infusion. **e** Normalized ripple rate across all animals and sessions (drug vehicle and CP,55–940 infusion; animal number = 10). Data are presented as mean ± SEM. **f** Normalized ripple count between baseline and infused vehicle or drug (two-sided Wilcoxon Signed-rank test, *****$p \leq 0.00001$, NA $p > 0.05$ (Baseline vs. VEH, $p = 0.5857$, animal number = 5), (Baseline vs. CP-55,940, $p = 1.7333e\text{-}6$, animal number = 5)). Data are presented as mean ± SEM.

generate SPW-Rs post drug is not due to an inability to sufficiently depolarize CA1 neurons.

Wavelet transform was additionally computed and averaged over all sessions to visualize optically induced SPW-Rs (Fig. 8e and Supplementary Fig. 8). Figure 8f shows the spectrum of baseline and after drug infusion at high optical pulses. The differences between baseline and post-drug infusion were larger at higher optical pulse intensity (Fig. 8g). The spectrum response to the high optical power in each session was min-max normalized and averaged (Fig. 8h; animal number = 4). The spectral difference between the baseline and CP-55,940 was computed and averaged, as shown in Fig. 8i. In all sessions, we observed an increase in the ripple rate during the optical stimulation. After infusion of CP-55,940, the ripple rate drastically decreased, as expected from Fig. 7. The optical stimulation after infusion did not recover the SPW-Rs. There was a significant decrease in normalized ripple rate after drug infusion compared to the baseline. We also observed a significant difference in the ripple rate during optical stimulations before and after CP-55,940 infusion. (Wilcoxon Signed-rank test, *****$p \leq 0.00001$, (Baseline vs. CP-55,940, $p = 1.8162e\text{-}5$, animal number = 4), (Optical Stim vs. Optical Stim after CP-55,940, $p = 8.2773e\text{-}6$, animal number = 4)). These data further support agonism of CB1Rs expressed by CA1 interneurons as the mechanism of agonist induced SPW-R suppression.

## Discussion

Here we fabricated T-DOpE probes via a scalable and low-cost thermal tapering method. The method enables high versatility and complexity in probe design, and the semi-automated connection method ensures scalability of the T-DOpE probes. Our in vivo study demonstrates promising future experiments with the T-DOpE probe's reliable and precise recordings with simultaneous optical, and chemical manipulations to understand complex neural circuitry. Here, we used T-DOpE probes to investigate the role of CA1 CB1R signaling in hippocampal activity including theta and SPW-Rs in behaving animals. Our results suggest a critical role for CB1R signaling in the ability for CA1 circuits to generate ripples.

Typically, it is desirable to achieve a thinner probe to minimize tissue response[82] (i.e., neuron-scale devices). The thermal tapering process accomplishes this goal by allowing us to make an ultrafine tip at the tissue interface while maintaining a sizable backend that is compatible with industrial-level soldering/bonding processes. For example, we can easily draw down mini-preforms to produce a 50 μm sensing tip with a backend diameter of 2 mm. Compared with clean-room microfabrication-enabled devices, our approach allows low-cost fabrication of flexible and multifunctional devices. Compared with fiber probes fabricated using TDPs, our T-DOpE devices enable dual-size end tips instead of a consistent diameter. In previous fiber probes, electrodes were manually connected by carefully scraping away insulated layers and then electrically connecting the exposed electrode to a pin. Microfluidic channels are connected by attaching polymer tubing to the fiber via a similar process as the electrode connection. Optical connection is done by attaching a ferrule to the back of the fiber, limiting it to one optical connection per device. This manual connection is a time-consuming and labor-intensive process that becomes increasingly difficult as the fiber gets thinner. In our T-DOpE probes, the relatively large size of the backend allows for a semi-automated connection process, reducing connection time, labor, and cost (Fig. 3a; Cost: <10 dollar per unit). In addition, the previous connection process only allows connections to the outer layers of the

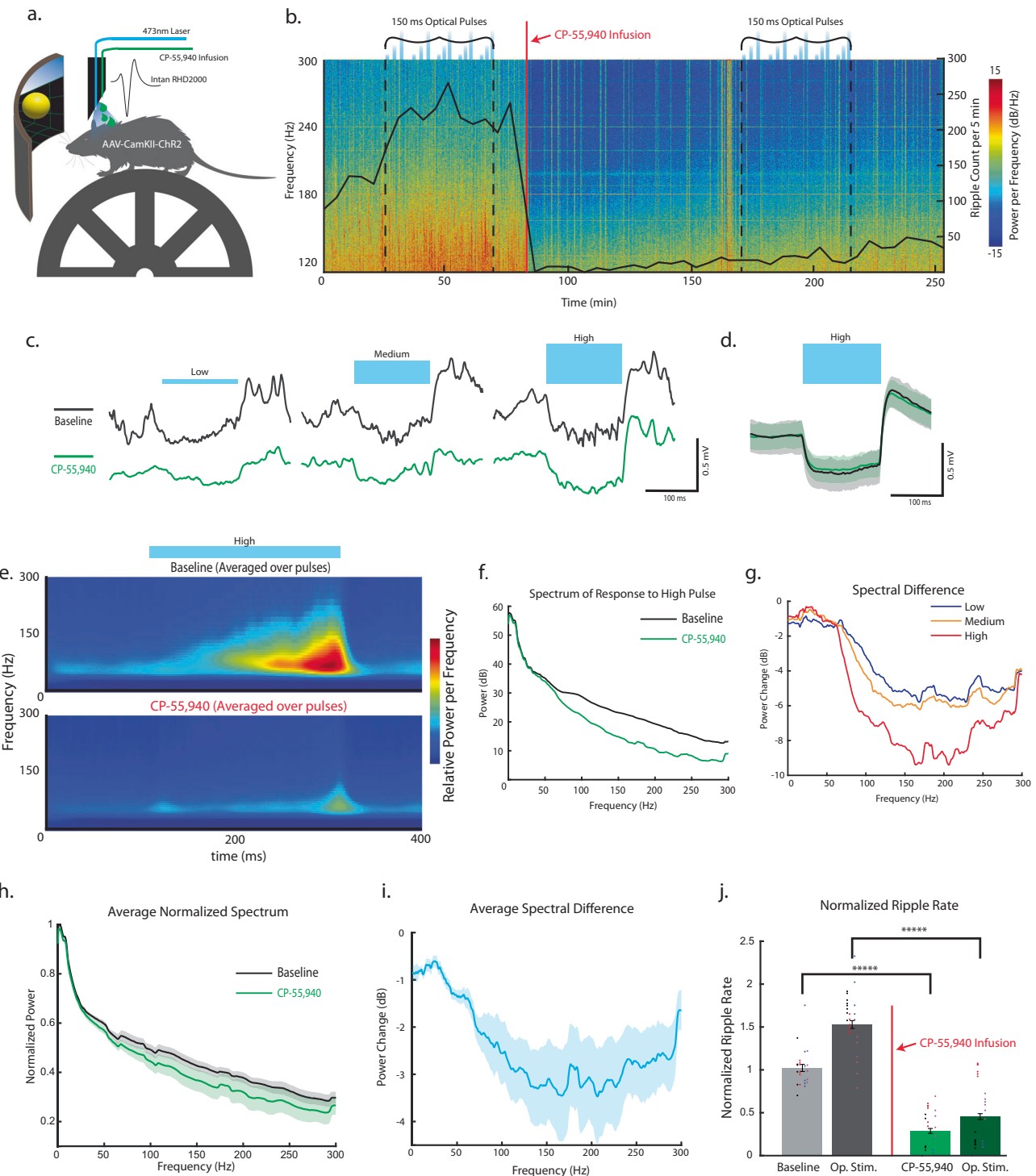

**Fig. 8 | Generation of SPW-Rs by optical stimulation of CA1 PYR is abolished by pharmacological activation of CA1 CB1Rs. a** Illustration of the experimental setup. A head-fixed AAV-CamKII-ChR2 mouse is mounted on the wheel. A virtual reality environment is presented on the screen to instigate running, and the virtual position is recorded. Local infusion, optical stimulation, and neural recording are achieved using the T-DOpE probe. **b** Power spectrogram of CP-55,940 infusion session on high frequency bins (100–300 Hz). Following 30 min of baseline, CA1 PYR are optical stimulated ($n = 400$ pulses of 150 ms at low, medium, and high power) over 40 min. CP-55,940 (mg kg$^{-1}$; 200 nL; 1 nLs$^{-1}$) is locally infused at the 80-minute mark. After the cells recover from the infusion, optical stimulation is repeated. **c** Representative response to low, medium, and high optical stimulations during the baseline and after CP-55,940 infusion. **d** Average neural response to the high optical stimulation during the baseline and after CP-55,940 infusion. Data are presented as mean ± SD. **e** Average Wavelet Transform of the activity in baseline and after drug infusion. **f** Spectrum of baseline and for CP-55,940 infusion during high pulses. **g** Spectral difference between the baseline and after CP-55,940 infusion for the optical pulses. **h** The averaged normalized spectrum (animal number = 4). Data are presented as mean ± SD. **i** The averaged spectral difference between the baseline and after CP-55,940 infusion for high optical pulses (animal number = 4). Data are presented as mean ± SD. **j** Normalized ripple rate during baseline and during optical stimulation. After the infusion, normalized ripple rate 30 min before and during optical stimulation. (two-sided Wilcoxon Signed-rank test, *****$p \leq 0.00001$, (Baseline vs. CP-55,940, $p = 1.8162e{-5}$, animal number = 4), (Optical Stim vs. Optical Stim after CP-55,940, $p = 8.2773e{-6}$, animal number = 4)). Data are presented as mean ± SEM.

device, and it is impossible to connect the channels near the center of the fiber without damaging outer channels. This restricts the complexity of the previous probes. Our connection process enables electrical, optical, and chemical modalities in the entirety of the fiber to be easily connectorized and also allows T-DOpE probes to be readily available for wide distribution. We are eager to disseminate these devices and will make them available to the neuroscience community upon reasonable request. Additionally, devices can be fabricated by neuroscience labs collaborating with engineering labs with relevant skills and customized equipment (see "Methods").

Using our device, we demonstrated precise optical and chemical modulations in vivo. We successfully identified the same monosynaptic connection between a putative pyramidal cell and an interneuron across 43 days which highlights the stability of long-term recording (Fig. 4c). The biocompatibility of the materials in T-DOpE probes (i.e., BiSn, PC, and PVDF) have been previously established[23,26]. By carefully varying the optical power, we were able to achieve different levels of manipulations of neural circuitry: increasing firing rate and optically inducing SPW-Rs (Figs. 8c, 7e). We also characterized the push and return of neurons after local infusion via electrophysiology (Fig. 5e). The slow local infusion allows us to study the pharmaceutical/chemical effects of a local neural circuitry. Depending on the type of chemical, volume, concentration, brain area, and duration of the infusion, the likely range is from 0.1–1 mm. The slanted cut of the tapered probe minimizes the tissue damage during implantation and enables depth-dependent recording along the device tip. With the T-DOpE probes, we always recorded the typical electrophysiological landmarks of hippocampus CA1, such as multi-unit activity, SPW-Rs, and theta oscillations (Fig. 4b). On top of the recording capability, T-DOpE probes enable optical stimulation for optogenetics and focal drug infusion for pharmacological intervention. The diameter of our probe is ~150 μm and is in between optical fiber (for optogenetics) and silicon probes. Our device is significantly smaller than a stainless steel cannula which is commonly used in intracranial drug infusion.

Hippocampal circuit activity is critical for episodic and spatial memory. Hippocampal theta (~6–10 Hz), gamma (~35–80 Hz), and sharp wave-ripple (~100–250 Hz) oscillations all contribute to mnemonic functions of the circuitry[33–37,73]. Cannabinoids impair memory and alter CA1 rhythms, suggesting a possible role for local CB1R in hippocampal activity and oscillations supporting memory[47,48]. However, CB1Rs are expressed in multiple hippocampal sub-regions (e.g., both CA1 and CA3) and cell types (e.g., excitatory pyramidal cells and inhibitory interneurons)[51,58–60,64,66], and thus the specific loci of action of cannabinoids on CA1 activity remain unknown. The development of our device has enabled us to further our understanding of the interaction of CB1R and hippocampal oscillations specifically through the focal infusion of agonist during behavior and optogenetic initiation of SPW-Rs.

Administration of CB1R agonists disrupts theta while keeping neuronal firing rate intact, through scrambling temporal coding by cell assemblies[47,48]. Thus while it is clear that cannabinoids disrupt neural circuits underlying theta, these studies were achieved through systemic administration of cannabinoids in rats. Until our study, it was unknown if cannabinoids were disrupting theta generated upstream of CA1, or within local circuitry. The medial septum, a proposed theta generator, has glutamatergic, cholinergic, and GABAergic connections to CA1[64]. These GABAergic and cholinergic neurons express CB1 receptors, likely inhibiting vesicle release at axon terminals[57,58]. One hypothesis is that systemic administration of cannabinoids may decrease theta by disrupting circuitry within the medial septum, which prevents effective theta drive in CA1. An alternative hypothesis is that cannabinoids bind to CB1 receptors expressed in CA1, either on CA1 neurons or terminals of inputs into CA1, altering precise timing of neurotransmitter release resulting in disrupted oscillations[48]. We demonstrated with our T-DOpE probe that local CA1 infusion of CP-55,940 was sufficient to disrupt theta (Fig. 6e), while time spent

running and average running speed did not change (Supplementary Table 2). We found there was no statistically significant relationship between theta power and running speed in our experiments (Supplementary Table 3), likely because navigation occurred in a virtual environment[83]. Focal CB1R activation in CA1 is sufficient to weaken the theta oscillation in behaving animals.

Cannabinoids inhibit SPW-Rs in rats[47], suggesting a potential mechanism underlying the memory impairment associated with cannabinoid use in humans. This effect was attributed to reduced glutamate release in CA1 from CA3 afferents[47,61,62], although CB1Rs are expressed in many locations within CA1, leaving the specific mechanism unknown. Importantly, activation of CB1Rs on medial septal inputs would be expected to increase and not decrease SPW-R rate (activation of the medial septum decreases SPW-R rate)[63]. Both CA3 and CA1 neurons express CB1Rs[51,58–60,64]. CB1Rs on CA3 axon terminals of the Schaffer collaterals have been shown to act to suppress glutamate release[51,60]. CA1 PYR express CB1Rs localized on their dendrites, activation has been shown to decrease post-synaptic excitability by increasing membrane conductances[67]. CB1Rs are highly enriched in CA1 interneurons where they inhibit GABA release[58,59]. Given the ambiguous relationship between locus of CB1R expression and SPW-R generation, it was unclear if cannabinoids were acting on excitatory drive from CA3 or local circuit interactions within CA1. Our device allows us to directly interrogate CA1 through local drug delivery and optogenetic stimulation. Our experiments show that CB1R activation is sufficient to disrupt SPW-Rs independent of CA3 synaptic transmission (Fig. 8e), even if the network is sufficiently activated by optogenetic depolarization of CA1 PYR (activating ChR2 expressing PYR is sufficient to generate CA1 ripples)[68]. (Fig. 8d). Thus while previous studies suggested that CB1R inhibit SPW-Rs by cutting off communication from other areas[61,62], our findings demonstrate the importance of physiological cannabinoid activity in the CA1 local circuit. Together, these findings reveal mechanisms by which cannabinoids disrupt specific hippocampal rhythms and suggest dysregulation of this tightly controlled system within the CA1 circuit may directly lead to memory impairment.

## Methods
All protocols and experiments were approved by the Virginia Tech (Blacksburg, VA, USA) Institutional Animal Care and Use Committee (IACUC).

### Preform fabrication
For this study, we fabricated preforms with three different designs corresponding to the 3 different devices used. Preform fabrication for the thermal drawing process (TDP) is an iteration of four steps: machining, inlaying, film wrapping, and consolidation. For all the preforms in this paper, consolidation is performed by heating the preform under vacuum at 190 °C. All polymer materials (films, tubes, rods, preforms and mini preforms) used during fabrication were baked under vacuum at 80 °C to ensure they are moisture free. For the 8 electrode, 1 microfluidic channel and 1 waveguide fiber preform, we start by rolling PVDF films (McMaster-Carr) onto a PC tube (McMaster-Carr). We then rolled PC films (Laminated Plastics) onto the preform and consolidated it. 8 grooves were machined into the consolidated preform and inlayed with BiSn alloy (Indium Corporation). Additional PC films were wrapped around the preform, which is then consolidated. The 8 electrode, 8 microfluidic channel and 4 waveguide fiber preform was fabricated by first milling 4 hollow channels into a solid PC rod. The preform was then wrapped with PC film and consolidated. 4 more hollow channels were machined, and the preform was again wrapped with PC film and consolidated. 12 more hollow channels were machined into the resulting preform, and were inlayed with 8 BiSn alloy strips (Indium Corporation) and 4 polymer waveguides (PC core, PMMA cladding). Finally, the preform was wrapped with additional PC film and consolidated. The 12 microfluidic channel, 8 waveguide fiber was fabricated in a very

similar manner. The only differences being 8 hollow channels were machined for the second layer of microfluidic channels, and there were 8 waveguides instead of 4 and no BiSn strips inlayed in the outermost layer of device feature elements.

## Mini-preform fabrication

Thermal drawing of mini-preforms was accomplished utilizing a specialized thermal draw tower which heats the preform in a custom-built furnace and pulls it down to mini-preforms via a capstan motor. The temperature of the furnace, the feeding rate, and the drawing speed control the final geometry of the mini-preform. The furnace is divided into top, middle and bottom sections that can be individually set to different temperatures. The top section preheats the preform, the middle section softens the preform and is where the fiber is drawn. The bottom section is where the fiber is cooled. For our fabrication process, the furnace's sections were set at 150 °C, 275 °C and 120 °C respectively (Note these may not be the actual preform temperatures as they are readouts of the temperature sensors mounted in the furnace). The drawn down fiber's diameter is maintained at 2 mm. The fiber is cut into 10 cm segments to form mini-preforms for thermal tapering process.

## Tapered device fabrication

Tapering was achieved using a custom thermal tapering setup built from optomechanical components (Thorlabs), a linear motor, a DC power supply, and a custom-built furnace consisting of a ceramic tube wrapped with nichrome heating wire and thermal insulation. The mini-preform was held in place and the alignment adjusted using opto-mechanical components. A DC power supply was then used to power a custom-designed furnace to 230 °C. Note this is the nominal temperature measured by a thermal coupler and may not be the mini-preform's actual temperature. Once the mini-preform was softened, it was pulled via a computer controlled linear motor (Zaber Technologies), resulting in a tapered structure. The speed of the motor and its travel distance can be utilized to adjust the tapered structure's geometry. To produce a neural probe which minimally damages the nearby tissue, the tapered structure was cut at an angle, resulting in two individual T-DOpE probes. This device can be built in an academic lab or engineering core facility with the custom equipment mentioned above and relevant skills.

## Assembly of multifunctional tapered microprobe

The connection of the T-DOpE probe was accomplished on a custom connection setup built with opto-mechanical components. Translation stages (Thorlabs) and a digital microscope (Linkmicro) were used to provide finer position control during connection. For electrical connection, the device backend was heated to 160 °C. This temperature is high enough to melt the BiSn electrodes but low enough to not damage the probe. 42AWG copper wires (Remington Industries) guided by a hypodermic needle were lowered into the melted electrodes. The device was then cooled to solidify the BiSN electrodes. Microfluidic connection was accomplished by inserting a custom drawn hollow (150 μm OD, 75 μm ID) PC tube into the microfluidic channels on the probe's back end. Thermally drawn 200 μm diameter Polymer optical waveguides (PC core, PMMA cladding) were polished (30-1 μm grit) and carefully coupled onto waveguides on the probe's connection end. To seal the microfluidic connection and keep the waveguide in place, UV resin (Piccassio) was applied to the entire probe backend and cured.

To properly interface with our recording setup, optical laser, and drug delivery systems, the probe is further fitted with adapter components. The probe's copper wires were soldered to either pin connectors (chronic implantation) or custom designed PCBs (acute implantation) which can be readily connected to Intan acquisition systems via PCBs purchased from NeuroNexus. The probe's waveguides were connected to Ø1.25 mm stainless steel ferrules (Thorlabs). The microfluidic tubes were connected to IDEX Health & Science fluidic components via UV resin. The IDEX components themselves were made compatible with our drug delivery system by a custom-made adapter.

## Electrochemical spectral impedance measurement

Impedance measurements are done via a potentiostat (Gamry Instruments). The measurements were performed by lowering the sensing end of a fully connected T-DOpE probe into phosphate-buffered saline (PBS, Thermo Fisher). A Pt wire (Basi) was used as a counter and reference electrode. Data acquisition was accomplished via Gamry's proprietary software.

## Headbar implantation

Mice are induced and maintained at a surgical plane of anesthesia with isoflurane while mounted in stereotax. Hair is removed and scalp is disinfected. Bupivicaine nerve block is injected once under the scalp. The scalp is removed with surgical scissors and the skull is cleaned and dried with 3% hydrogen peroxide, followed by application of the sterile dental adhesive Optibond (Kerr Dental), which is cured with blue light. A < 0.2 mm burr hole is made above the right cerebellum, and a stainless-steel wire is inserted between the skull and the brain, parallel to the brain surface, then the wire is affixed to the skull with sterile dental acrylic. This wire is connected to the ground of the system. A titanium headplate (2 cm long, ~1 gram) is positioned above lambda, parallel to skull, and permanently fixed in place with sterile dental acrylic.

## Craniotomy

Mice are induced and maintained at a surgical plane of anesthesia with isoflurane while mounted in the stereotax. A 0.5–1.0 mm burr hole (using dental drill with 0.2 mm burr bit) is made above the hippocampus and the dura is removed. Biocompatible silicon elastomer (Kwik-cast; World Precision Instruments) is applied to the burr hole.

## AAV injection

Mice are induced and maintained at a surgical plane of anesthesia with isoflurane while mounted in stereotax. Hair is removed and scalp is disinfected. Bupivicaine nerve block is injected once under the scalp. A < 0.2 mm burr hole is made above the hippocampus (mm from bregma: −1.8, lateral: 1.5). Glass pipette containing AAV5-CaMKIIa-hChR2(H134R)-EYFP (UNC Gene Therapy Center – Vector Core) is lowered into CA1 (mm from surface: −1.2). 100 nL of AAV (titer: $4.1 \times 10^{12}$ GC/mL) is injected into tissue ($1 \, nLs^{-1}$) using microinjector syringe pump (WPI: MICRO2T & 504127). Glass pipette is left for 5 min for virus to diffuse, then removed from brain. Craniotomy is covered using biocompatible silicon elastomer (Kwik-SIL; World Precision Instruments) and scalp is closed with Vetbond (3 M).

## Drug preparation

CP-55,940 stock was prepared in ethanol at a concentration of 1.68 mg/mL. CP-55,940 injection solution was a mixture of 1:1:18 ethanol, solubilizer, and saline for a final drug concentration of 84 μg/mL. 200 nL of solution was delivered at a rate of $1 \, nLs^{-1}$ directly into CA1. For a total delivery of an estimated 16.8 ng or 44.6 picomols. Drug vehicle solution was a mixture of 1:1:18 ethanol, solubilizer, and saline. 200 nL of vehicle solution was delivered at a rate of $1 \, nL \, s^{-1}$ directly into CA1. We used a standard precision injection apparatus (NanoFil Syringe and UMP-3 Syringe pump, World Precision Instruments) for all tests and experiment.

## In vivo recording

All mice used in experiments were derived from crossing C57BL/6 J (JAX #000664) x FVB/NJ (JAX #001800). During the time of experiments, mice were between the ages of 16–30 weeks old. Mice of both sex were used in all experiments. Mice, trained and habituated to head-fixed navigation, are placed in the head fixation apparatus, and the

T-DOpE probe is lowered through the craniotomy into the brain until SPW-Rs are recognized and left in place for 30–45 min until tissue is relaxed. The amplified neural signals are then recorded with RHD2000 system (Intan Technologies LLC). For chronic recordings, probe was fixed to the skull with dental cement, after the probe was lowered into CA1 and SPW-Rs were identified.

## Optical stimulation
The T-DOpE probe's optical ferrule was coupled to a diode-pumped solid-state (DPSS) laser (Laserglow Tehcnologies, 100 mW maximum power, wavelength = 473 nm.) through a mating sleeve (Thorlabs). We calibrated the optical output of our probe for each optical session. The optically evoked activity was closely monitored to meet the desired power output. The outputs of the optical power varied between 50–700 µW. We calculated the optical power density to be 3–47 mW/mm$^2$. For investigating the effect of CP-55,940, 400, 150 ms optical pulses at low, medium, and high power were delivered. Time in between the power levels were set to 1 s (one set), and time in between stimulation sets was 5 s long.

## Data analysis
Data analysis was carried out with Matlab (The Mathworks) and custom scripts were used to analyze the extracellular recording. Due to the nature of the experiment where the researcher had to prepare the drug immediately before the experiment, they were not blinded to the manipulation conditions. However, analysis pipelines including ripple detection and theta power detection were consistent across animals and were not manipulated for experimental condition. For each session, the LFP power analysis was performed on the electrode right above the electrode detecting highest power of SPW-R. The data was low-pass filtered and then downsampled from 30 kHz to 1250 Hz. Spectrograms were computed to visually analyze the sessions in both time and frequency domain. The calculations of spectrograms were computed using hamming window of 5 s and overlaps of 2.5 s. The calculation of power spectral density was performed by using the multi-taper estimate method. SPW-Rs were detected using the LFP data. The LFP was first spectrally filtered from 100 to 250 Hz and threshold filtered to 1.5–3.0 of the standard deviation of the whole data. The events that exceed 40–50 ms were counted towards the SPW-R events. The SPW-R was visually inspected and compared with the raw recording. The threshold and the duration of the SPW-Rs were manually adjusted to minimize the error in detection. The running epochs were computed by taking the derivative of the virtual position of the mouse (>2 cm/s and >1 s running epochs). The power spectra of the running epoch were also computed using the multi-taper estimate method. The spectra were also minmax normalized for the comparison over all the session (Fig. 6c, d). Since the peak of the theta power varied over the animals and sessions, we computed the area under the curve (6–11 Hz) for the comparison between baseline and vehicle or drug (Fig. 4e). The average running speed were computed for each running epoch. The average running speed pre and post infusion was also computed to compare running behavior before and after drug infusion. The percent time spent running was computed for all sessions. The Shpiro-Wilk normality test was adapted to check for normality and the paired t-test was used to observe the running behavior pre and post infusion. Linear mixed-effects model was used to characterize the relation between the running speed and the theta power. Spectra of the optically induced activity were computed using the multi-taper estimate. To maintain the temporal information, the Wavelet Transform was computed and averaged. Spike sorting was performed using Kilosort1, followed by manual refinement using phy[84,85]. The putative cell type identification and cluster quality analysis was completed using Cell Explorer[74]. The cross-correlogram and auto-correlogram were normalized for visual observations. The two comparison groups were tested with Shapiro-Wilk normality test to check if the groups are normally distributed. If the two groups were normally distributed, we used the paired two-sided T-test. For those that are not normally distributed, the experimental designs with two comparison groups were analyzed by two-sided Wilcoxon signed rank tests. The difference between the two groups were considered statistically significant if the p-value is less than 0.05. All tests were performed using Matlab code.

## Reporting summary
Further information on research design is available in the Nature Portfolio Reporting Summary linked to this article.

## Data availability
The data that support the findings of this study are available from the corresponding author upon request. Source data are provided with this paper.

## Code availability
Much of the code used for this study was adapted from the buzcode repository (https://github.com/buzsakilab/buzcode). The other MATLAB scripts for analysis are available from the corresponding author upon request.

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

## Acknowledgements

We thank Matthew Buczynski and Cristina Miliano for advising us with the drug delivery of cannabinoids in vivo. X.J. gratefully acknowledges funding support from the National Institute of Health (R01NS123069, R21EY033080, R56AG077720) and National Science Foundation (ECCS-1847436). DFE gratefully acknowledges funding support from The Simons Foundation and The Whitehall Foundation.

## Author contributions

J.K., H.H., E.G., D.E., and X.J. designed the study. H.H. developed the tapering process. J.K. and H.H semi-automated the device connections. H.H. fabricated the T-DOpE probes used in the in vivo experiments. J.K. and E.G. carried out the animal surgeries. J.K. and H.H. optimized the electrophysiological recording, optical stimulation, and drug delivery setup. J.K. and E.G. executed infusion experiments. J.K., H.H., and E.G. conducted simultaneous optical stimulation and drug infusion experiment. J.K., H.H., E.G., and K.A. analyzed electrophysiology data. All the authors contributed to the writing of the manuscript.

## Competing interests

The authors declare no competing interests.
