## [Peer Review File · Nature Communications]

T-DOpE probes reveal sensitivity of hippocampal oscillations to cannabinoids in behaving miceREVIEWER COMMENTS

Reviewer #1 (Remarks to the Author):

The authors demonstrate technological improvements in practical points and resolution of neural probes that are based on thermal fiber drawing fabrication approach, by conceiving the additional thermal tapering process in the approach. High-precision multimodal functionality of the sharp neural probe allowed study to investigate CB1Rs local to the Hippocampal CA1. They argue that their scientific finding on the effect of region-specific CB1Rs on SPW-Rs is the main novelty of the paper. Organized systematic demonstrations on technological improvements of devices with control experiments well support the scientific advancement.

Major comments

1. It would be useful for readers to add a reference which can provide detailed mechanisms of the thermal tapering process which is also used in other fields. An example reference: Korposh, Sergiy, et al. "Tapered optical fibre sensors: Current trends and future perspectives." *Sensors* 19.10 (2019): 2294.
2. Although fine resolution of neural probe is repeated argued, quantitative information showing resolution of multimodal functions is not provided. For example, what is the minimum size of microchannel achievable without any defects like localized collapse during the thermal tapering process.
3. It was interesting to see the results regarding the general technical issue in drug infusion studies, which is demonstrated in Fig.4e. Was the hypothesized physical cell displacement being commonly experimentally noticeable with conventional neural probes? A reference regarding the information will be useful for the readership if there is. If there are improvements in the issue for the proposed device in comparison to the conventional, please provide explanations.
4. What is the meaning of the speed data in the upper panel of Fig. 5b and Fig. 6a? Explanations of the data are not provided in the results.

Minor comments

1. Missing full word for abbreviation of PYR in line 334. The full word was in line 425.
2. For the demonstration on flexibility in Fig.3C, quantitative information of bending stiffness would be useful for understanding the advantage of the device.

Reviewer #2 (Remarks to the Author):

Kim et al describe a novel device that fills an important need in neuroscience for multimodal tools that can simultaneously chemically and optically manipulate the nervous system while recording electrical neural activity. Using this new device, the authors present scientific findings showing that local cannabinoids affect theta and SWR activity in CA1, however the significance of these scientific findings is unclear. While the device is novel and potentially impactful, more rigorous device characterization is required, including comparison to gold-standard techniques in the field. The scientific findings require a clear rationale and hypotheses. For both the device characterization and scientific findings replication in more animals and more rigorous statistical tests are needed. These points are elaborated below:

Major concerns

- To give potential users a better idea of how they could incorporate this device into their experimental plans, the authors should explain the logistics needed to build it. How hard is this device to build? What equipment or skills are needed? Is this something a lab would likely build themselves or purchase? How long does it take to build?
- Some basic device characterization is missing. For example
 - o What is the light power emitted by the waveguides? Over what area?
 - o Over what area is drug infused over how long? I expect this could be adjusted based on the exact

infusion protocol but what is the range of possible areas affected.

o What is the typical yield of single units per recording? This should be replicated in at least 5 mice. How many putative pyramidal cells and interneurons were clustered?

o What is the clustering quality over many cells? what is the distribution of single unit clustering metrics?

o How reliable are each of these key features (light power, drug area covered, single unit yield, cluster quality) replicated over many experiments?

- This device should be compared to the current gold standards in the field. For all of the above questions, how does this device compare to current standard devices. For example, how does the single unit yield compare to current silicone probes and tetrodes? Furthermore, how does the diameter of this device compare to current silicone probes, tetrodes, optical fibers, and infusion devices. How does the impedance of these electrodes compare to tetrodes and silicone probed?

- What is the size of each channel, waveguide, and electrode in Fig 2D?

- What is the effect of the optical stimulation on single neurons? The authors should confirm that it is possible to precisely control spike timing of single units, not just alter LFP, as optogenetic manipulation of single units is the common standard in the field.

- As written, the manuscript dedicates a lot of space to scientific experiments (three figures) but does not provide strong rationale for why these experiments in particular are important. In many ways these findings replicate and confirm prior work that has shown systemic manipulation of cannabinoids affects theta and SWRs or local infusion in anesthetized animals affects SWRs. While this study manipulates cannabinoids locally in awake behaving animals, it is not clear why that manipulation is important or how these results change the way we think about the role of cannabinoids affect hippocampal activity. In principle, I agree it is important to differentiate between local or non-local mechanisms, but I do not understand the significance of local vs non-local cannabinoid actions in hippocampus. I left thinking, what have I learned? The authors could resolve this in different ways. One option would be to minimize the scientific findings and explain this is simply a demonstration of the technology with the aim of replicating prior work with better localization in awake animals. Another option would be to elaborate more on the scientific background, rationale, and hypotheses that guide these experiments. Why do we care about cannabinoids effects on hippocampal activity? Why do we care if the effect is local or not? What are the open questions in this area and why are they important? How do these findings change our mental model of how hippocampus functions?

- Several figures have n of 2,3, or 4 animals. This is too few to draw conclusions. Studies should be replicated in at least 5 animals.

- Why is the relationship between speed and theta power different between the baseline conditions in Fig 5f and 5g? These are similar experiments and the relationship should be similar.

- The statistical approach, primarily student's T-tests, ignores interdependence in data recorded from the same animal. When data recorded from the same animal is treated as independent statistical power is inflated. A more appropriate approach would be linear mixed models. Furthermore, the authors should not assume data is normally distributed without testing whether that is the case.

Minor concerns

- The text on the panels in Figure 2d is really hard to read

- The saline and vehicle infusion controls are very important and should be in the main figures

- Were experimenters blind to manipulation conditions during experiments and analysis?

Reviewer #3 (Remarks to the Author):

The authors devised a method to fabricate flexible, multifunctional probes (T-DOpE) with electrodes, waveguides, and microfluidic channels in one device via a thermal drawing process. The diameter of the probe tapers down from 2 mm, allowing for easy backend connections, down to 150 μm which enables high spatial specificity. T-DOpE was used to investigate the mechanism of action of CP-55,940, a cannabinoid, in the CA1 region in the hippocampus. First, the authors showed that focal CA1 cannabinoid receptor (CB1R) agonism, not systemic agonism, is sufficient to reduce theta oscillations

in CA1 and to reduce short-wave ripple (SPW-R) rates. Using mice expressing channel rhodopsin (ChR2), which optogenetically induces SPW-R, in CA1, the authors addressed the question of whether the reason SPW-R rate decreases after CP-55,940 infusion is due to CB1R agonism at the synaptic terminals of CA3 inputs and/or hippocampal interneurons. SPW-R was locally induced in CA1 via optical stimulation of ChR2. It was found that SPW-R decreased as a result of CP-55,940 infusion, confirming that agonism of CB1R by CP-55,940 in CA1 interneurons is responsible for SPW-R suppression. The multi-functional probe and its use for a specific neuroscience study is certainly very interesting. With that being said, the proposed device seems to be a relatively trivial step on top of the thermal drawing process that has been reported previously to fabricate multifunctional probes (Canales et al. *Nat Biotechnol*, 2015). So it is not clear from the paper where is the innovation of this method. Also, not much characterization is done on the device either. Furthermore, since the neuroscience conducted in this study is very niche, the broader implications of the findings are not clear. Below are some specific questions and suggestions.

Questions:

1. What are the optical properties of the probe like? Transmission spectra would be useful.
2. What are the mechanical properties of the probe like? What is the stiffness? What is the ultimate flexural strength?
3. Depending on the mechanical properties, the probe could break if handled in an incorrect manner. Is this a concern?
4. For Figures 5f and 5g where the theta power is plotted against the speed, is the difference between the two linear regressions statistically significant? A two-way ANOVA or a statistics test of the author's choice would be helpful. Also, what is the correlation strength for the linear regressions? Depending on these quantities, the data presented in Figure 5h could be misleading.
5. What are the broader implications of the findings? How does it help the field advance its understanding of the hippocampus and memory? Do the findings have the potential to lead to practical applications?

Suggested Revisions/Comments:

- The introduction section begins with a technology-level approach, but most of the manuscript is centered around a fundamental neuroscience question, making the logic of the manuscript confusing. The manuscript should be re-structured so that the most important aspect of the work is emphasized.
- The discussion section is too long. A significant portion of the discussion section should go in the introduction section. Some of the content in the discussion section will help contextualize the motivation of the study if it is moved to the introduction section.

Response to reviewers' comments:

We would like to thank all the reviewers for their deep and thorough review of our manuscript, which helped us improve and clarify the manuscript. Reviewers' comments are in *italic*, and the authors' responses are in black. Main text changes are highlighted in red.

Reviewer #1 (Remarks to the Author):

The authors demonstrate technological improvements in practical points and resolution of neural probes that are based on thermal fiber drawing fabrication approach, by conceiving the additional thermal tapering process in the approach. High-precision multimodal functionality of the sharp neural probe allowed study to investigate CB1Rs local to the Hippocampal CA1. They argue that their scientific finding on the effect of region-specific CB1Rs on SPW-Rs is the main novelty of the paper. Organized systematic demonstrations on technological improvements of devices with control experiments well support the scientific advancement.

We thank the reviewer for the positive assessment of our manuscript. Here we provide the point-to-point response to each comment as shown below. We hope the reviewer finds that these additional results and discussion significantly strengthen our manuscript.

Major comments

*1. It would be useful for readers to add a reference which can provide detailed mechanisms of the thermal tapering process which is also used in other fields. An example reference: Korposh, Sergiy, et al. "Tapered optical fibre sensors: Current trends and future perspectives." *Sensors* 19.10 (2019): 2294.*

We thank the reviewer's suggestion on adding a reference for the thermal tapering process. We have added the reference and a sentence to the results section to provide detailed mechanisms of our thermal tapering process adapted from those used in other fields.

The process is adapted from the fabrication method of heating glass pipettes and tapered silica optical fibers.⁷² Here, instead of using a single material (silica) as in the traditional tapering approaches, we develop a multi-material thermal tapering approach to co-draw multiple functional components inside a polymer matrix.

72. Korposh, S., James, S.W., Lee, S.-W. & Tatam, R.P. Tapered Optical Fibre Sensors: Current Trends and Future Perspectives. *Sensors* 19, 2294 (2019).

2. Although fine resolution of neural probe is repeated argued, quantitative information showing resolution of multimodal functions is not provided. For example, what is the minimum size of microchannel achievable without any defects like localized collapse during the thermal tapering process.

We appreciate the reviewer's suggestion and have added more results on the quantitative assessment of our probes.

The optical transmission spectrum of design 1 is included in Supplementary fig. 5, which shows the optical transmission across the visible wavelength range (400-750nm).

By varying illumination intensity (from 3mW/mm² to 47mW/mm²) we were able to produce a range of changes in the spike rate (fig. 5ab). These levels are within the typical range for optogenetics.

The minimal diameter of tapered multifunctional fibers is 50 μm (Supplementary fig. 1).

Drug delivery channels can be as small as 20 μm without collapsing (Supplementary fig. 2). Here we used 40 μm diameter microfluidic channels because the drug we injected is of high viscosity which we found incompatible with the 20 μm sizes. A new figure in the supplement has been added demonstrating the various diameters achievable with thermal tapering (Supplementary fig. 1). We additionally included the optical transmission spectrum measurement of our device in the supplementary. The following text has been added to the main text and the method section:

By varying the temperature and the pulling speed thicker (200 μm in diameter) or thinner tapered fiber (50 μm in diameter) can be achieved, as shown in **supplementary fig. 1**. To meet the impedance requirement for high-quality neural recording, the electrodes were designed to be at least 25 μm in diameter. Drug delivery channels can be as small as 20 μm without collapsing (**Supplementary fig. 2**). However, the pressure required to push fluid through a microfluidic channel is proportional to the channel radius⁴ and fluid viscosity. For this study in which we delivered vehicle and CP-55,940 which has a high viscosity, we chose drug channels with a diameter of ~ 40 μm to ensure stable and reliable drug delivery.

Supplementary Figure 1: Microscope image of three fibers with various diameters (200 μm , 100 μm , 50 μm) fabricated using thermal tapering.

Supplementary Figure 2: Cross section of a probe design with 20 μm microfluidic channel.

We calculated the light power density to be 3 - 47 mW/mm².

By regulating optical stimulation power, we were able to modulate individual unit spike rate.

a. Example trace of wideband extracellular response to optical stimulation. **b.** The firing rate of sorted cells during optical stimulation and the normalized firing rate across all sessions.

The transmission spectrum of the optical waveguide is included in the supplementary fig. 5.

Supplementary Figure 5: Transmission Spectrum of the optical waveguide

3. It was interesting to see the results regarding the general technical issue in drug infusion studies, which is demonstrated in Fig.4e. Was the hypothesized physical cell displacement being commonly experimentally noticeable with conventional neural probes? A reference regarding the information will be useful for the readership if there is. If there are improvements in the issue for the proposed device in comparison to the conventional, please provide explanations.

We thank the reviewer for the comments on the physical displacement of the cell. Previous studies using specialized devices to deliver drug at the same location of neuronal recording have reported similar effects (Shin et al., Lab on a Chip. 2015, Shin et al., Nature Communications. 2019, Yoon et al., Nature Communications. 2022). In our manuscript, we additionally studied for the first time, dynamic changes in the amplitude and firing rate of the spikes during infusion. We also newly provide a detailed comparison between saline, vehicle, and drug infusion to validate that the neural activity change indeed comes from the drug itself instead of cell displacement caused by infusion. As an aside, the physical cell displacement further validates that the drug is successfully delivered. We have conducted additional experiments in this revision (5 saline, 5 drug vehicle, and 9 drug infusions) which is consistent with our findings.

We decided to divide our figure 4 into two figures to address this phenomenon more.

To validate that the transient silencing of cells is a result of the physical cell displacements^{72, 78, 79} instead of pharmacological influence, we performed a detailed comparison of the dynamic spike recording during the infusion of saline, vehicle, and drug with vehicle. We observed transient absences of spikes after infusion of either saline, vehicle, or drug+vehicle (**fig. 5d, supplementary fig. 7**).

72. Korposh, S., James, S.W., Lee, S.-W. & Tatam, R.P. Tapered Optical Fibre Sensors: Current Trends and Future Perspectives. *Sensors* **19**, 2294 (2019).
78. Yoon, Y. et al. Neural probe system for behavioral neuropharmacology by bi-directional wireless drug delivery and electrophysiology in socially interacting mice. *Nature Communications* **13**, 5521 (2022).
79. Shin, H. et al. Neural probes with multi-drug delivery capability. *Lab on a Chip* **15**, 3730-3737 (2015).

d. Normalized firing rate of sorted cells in saline, vehicle, and drug+vehicle infusion. Note shortly after the infusion, the firing rate of all cells drastically decrease not due to pharmacological influence but to the physical displacement of the cells from the recording sites. **e.** Spike amplitude of an interneuron and a pyramidal cell to demonstrate the recovery after infusion (200nL ; 1nL s^{-1}). Given the device tip contains both the microfluidic channel and recording sites ($< 20\ \mu\text{m}$ in distance), the cells are pushed away and return after some period of time. **f.** Average spike waveforms of the two cells before and after infusion. **g.** Autocorrelation of the two cells before and after infusion.

4. What is the meaning of the speed data in the upper panel of Fig. 5b and Fig. 6a? Explanations of the data are not provided in the results.

We thank the reviewer for the comments on the speed data in the recording. While the mice are running, the assemblies of place cells are active in synchrony to produce the theta oscillation (6~11 Hz). We wanted to compare the theta oscillation before and after CP-55,940 focal infusion, as previous studies show systemic infusion of cannabinoid weakens the theta oscillation, and at excessively high doses eliminates ambulation. The speed of the data is recorded to determine the running epochs of the mice, and to demonstrate that the dosage we administer does not affect the gross motor activity of the mice (i.e. they continue running) following infusion.

We have added the following text to the results section to provide an explanation of the running data. In addition, we added the running velocity data to supplementary table 2 to show the drug infusion does not eliminate running behavior:

While mice run assemblies of place cells are dynamically synchronized with the theta oscillation (6~11 Hz). The speed data is used to determine the running epochs of the mice, and to demonstrate that mice continue running following infusion. The average speed of each session is shown in **supplementary table 2**.

Linear mixed estimate model fitted by using fitlme (Matlab)
 Power of theta as function of intercept and running speed grouped by infusion conditions
 Mean velocity pre and post infusion (1 hour each)

Infusion type	Mouse ID	Mouse Session	P-value for slopes	P-value for intercept	Avg. vel. & std preinfusion (cm/s)	Avg. vel. & std preinfusion (cm/s)
Drug Vehicle	m400	m400_221219	0.56272	8.454e-66	0.22; 0.39	0.28; 0.51
	m401	m401_221224	0.53787	1.4772e-34	0.39; 0.70	0.40; 0.78
	m402	m402_221212	0.19012	1.4671e-105	0.87; 1.62	0.76; 1.70
	m442	m442_231006	0.629	6.1918e-32	0.54; 1.3	0.51; 1.52
CP-55,940	m402	m402_220926	0.37642	3.176e-104	0.81; 1.67	0.57; 1.29
	m403	m403_221011	0.05195	1.5538e-58	0.38; 1.01	0.34; 0.75
	m404	m404_221018	0.69183	4.0685e-109	0.42; 2.20	0.49; 1.73
	m441	m441_231005	0.10348	1.0033e-38	1.27; 3.35	1.25; 3.01
	m442	m442_231010	0.39025	1.6338e-24	1.07; 2.61	1.00; 2.92

Supplementary Table 2: Linear mixed-estimate model of the theta power, velocity of the mice, and the infusion condition. The average velocity before and after the infusion of vehicle or CP-55,940.

Minor comments

1. *Missing full word for abbreviation of PYR in line 334. The full word was in line 425.*

We thank the reviewer for the comment. We have added the full word for abbreviation of PYR.

2. *For the demonstration on flexibility in Fig.3C, quantitative information of bending stiffness would be useful for understanding the advantage of the device.*

We thank the reviewer for the suggestion on adding quantitative information of bending stiffness of our device. Following the reviewer's suggestion, we conducted additional experiments to test the bending stiffness using a dynamic mechanical analyzer (DMA Q800 from TA instrument). We measured the bending stiffness of the three designs and a stainless steel wire (same diameter) and found that our probes are much more flexible than stainless-steel wire. Critically, as outlined in the results section, the impedance of our electrodes before and after bending remains similar.

We have added the bending stiffness measurement in the supplementary information and the following text to the main text:

Using a dynamic mechanical analyzer, we measured the stiffness of the three designs and a stainless steel wire, included in **supplementary fig. 4**. Note that all designs have a much lower stiffness than the stainless steel wire with the same diameter.

Supplementary Figure 4: Stiffness vs Frequency measurement using a dynamic mechanical analyzer

Reviewer #2 (Remarks to the Author):

Kim et al describe a novel device that fills an important need in neuroscience for multimodal tools that can simultaneously chemically and optically manipulate the nervous system while recording electrical neural activity. Using this new device, the authors present scientific findings showing that local cannabinoids affect theta and SWR activity in CA1, however the significance of these scientific findings is unclear. While the device is novel and potentially impactful, more rigorous device characterization is required, including comparison to gold-standard techniques in the field. The scientific findings require a clear rationale and hypotheses. For both the device characterization and scientific findings replication in more animals and more rigorous statistical tests are needed. These points are elaborated below:

We thank the reviewer for the positive assessment of our newly developed probe and the constructive comments, which helped to improve our manuscript. We have provided more detailed discussion on the significance of the scientific findings. We have also conducted comprehensive experiments on the device characterization including comparison with gold-standard techniques in the field. We have increased the animal number in each experiment (5 saline, 5 vehicle, 5 CP-55,940, and 4 CP-55,940+opto) and provided more rigorous statistical tests. We hope the reviewer finds that these additional results and discussion significantly strengthen our manuscript.

Major concerns

1. To give potential users a better idea of how they could incorporate this device into their experimental plans, the authors should explain the logistics needed to build it. How hard is this device to build? What equipment or skills are needed? Is this something a lab would likely build themselves or purchase? How long does it take to build?

We thank the reviewer for these great questions. A neuroscience lab would likely need to collaborate with an engineering lab to fabricate the customizable device. The engineering lab would need to be equipped with a fiber drawing setup, a tapering setup, and optical and electrical testing equipment. The skills and expertise required to fabricate T-DOpE probes depend on the complexity of the design and material. The user needs to be familiar with the properties of the materials in the probe and be able to adjust the temperature and pulling speed accordingly. The custom thermal tapering setup requires equipment that is specialized but can be assembled with off the shelf components. We are eager to disseminate these devices which will be available upon reasonable request after publication.

To ensure future readers are made clearly aware of the fabrication steps and required equipment, we have made the following modifications to our methods section, where the fabrication steps and equipment necessary can be found.

Thermal drawing of mini-preforms was accomplished utilizing a specialized thermal draw tower which heats the preform in a custom-built furnace and pulls it down to mini-preforms via a capstan motor. The temperature of the furnace, the feeding rate, and the drawing speed control the final geometry of the mini-preform.

Tapering was achieved using a custom thermal tapering setup built from optomechanical components (Thorlabs), a linear motor, a DC power supply, and a custom-built furnace consisting of a ceramic tube wrapped with nichrome heating wire and thermal insulation.

2. Some basic device characterization is missing. For example

- What is the light power emitted by the waveguides? Over what area?*

The outputs of the optical power varied between 50 – 700 μW depending on the desired neural responses over an area of 0.00157 mm^2 . From this, we calculated the optical power density to be 3 – 47 mW/mm^2 .

We added the following sentence to the methods section.

We calculated the optical power density to be 3 – 47 mW/mm^2 .

- Over what area is drug infused over how long? I expect this could be adjusted based on the exact infusion protocol but what is the range of possible areas affected.*

The reviewer is correct that the spread of drug and area affected is important and can be adjusted depending on the exact infusion protocol. To the best of our quantification, our infusion of 200nL of the lipophilic drug CP-55,940 in vehicle in

the pyramidal layer of CA1 is restricted to <300 μm range over 200 seconds. Importantly, the design of infusion protocol was informed by previous works that demonstrated local delivery: intrahippocampal delivery of CP-55,940 (Wise et al., *Neuropsychopharmacology*. 2009, Robbe et al., *Nature Neuroscience*. 2006) (500 μl through metal cannula), DAPI infusion protocol through drug delivery probes (Shin et al., *Lab on a Chip*. 2015), and our AAV injection protocol as well as many others that demonstrate restricted expression to CA1 (Dombeck et al., *Nature Neuroscience*. 2010, Trouche et al., *Nature Neuroscience* 2016, Kampasi et al., *Microsystems & Nanoengineering*. 2018, Mau et al., *Current Biology*. 2018, Opalka and Wang, *Learning and Memory*. 2020, Dong et al., *Nature Communications*. 2021, Wirtshafter and Disterhoft, *Journal of Neuroscience*. 2022, Ding et al., *eLife*. 2022, Schuette et al., *Journal of Neuroscience*. 2022). We conservatively infused 200nL of volume at a rate 1nL/s. Importantly, given the lipophilic nature of CP-55,940 we anticipate the spread to be less than any of these water-soluble infusions. Depending on the type of chemical, volume, concentration, brain area and duration of the infusion, the likely range is from 0.1-1mm. For large volumes one would choose to use a different device such as a stainless steel cannula for drug delivery.

- *What is the typical yield of single units per recording? This should be replicated in at least 5 mice. How many putative pyramidal cells and interneurons were clustered?*

We have made a table of our recorded sessions with the number of putative single units and cluster quality. We conducted additional experiments and analysis and summarized the recording results from 15 mice. In some experiments single units were not able to be sorted, likely due to the distance between electrode sites.

We have included the following table in our supplementary and added the following text in the results section.

Though multi-unit activity (MUA) and power in the spike frequency band is observed in all recordings, we used Kilosort and manual curation using Phy to detect sortable units in 13 out of 19 acute sessions (n=15 animals). Detected units ranged from 1-7 units (median = 3) with 57.9% of units identified as putative pyramidal cells, and 42.1% as putative interneurons. Validated spike sorting metrics⁷⁴ of Isolation Distance ($\mu=26.9$, $\sigma=9.7$), and percentage of spikes that violate the inter-spike interval (<2ms, $\mu=0.53\%$, $\sigma=0.61\%$) are reported.

74. Petersen, P.C., Siegle, J.H., Steinmetz, N.A., Mahallati, S. & Buzsáki, G. CellExplorer: A framework for visualizing and characterizing single neurons. *Neuron* **109**, 3594-3608.e3592 (2021).

Treatment Group	Mouse ID	Session ID	# of Units	Putative Cell type	Isolation Distance	% ISI Violation
Saline	m449	m449_230921	3 units	PYR	26.3	0.12
				INT	21.0	0.09
				INT	27.6	0.01
	m450	m450_230921	1 unit	INT	28.2	0.27
	m451	m451_230922	1 unit	INT	38.4	0.03
m452	m452_290922	1 unit	PYR	19.0	3.31	
Drug Vehicle	m400	m400_221219	MUA only			
	m401	m401_221224	MUA only			
	m402	m402_221209	5 units	PYR	18.7	0.51
				PYR	25.9	0.21
				PYR	26.6	2.04
				PYR	14.4	0.72
				INT	28.7	0.24
		m402_221212	2 units	PYR	23.8	0.30
				INT	22.9	1.45
	m441	m441_231002	1 unit	PYR	21.5	2.16
m442	m442_231006	1 unit	PYR	18.7	0.15	
CP-55,940	m402	m402_220926	3 units	PYR	44.1	0.01
				PYR	4.9	0.62
				INT	1.01	0.95
	m403	m403_221011	3 units	PYR	33.2	0.21
				INT	35.0	1.82
				INT	38.7	0.47
	m404	m404_221018	MUA only			
	m441	m441_231005	MUA only			
	m442	m442_231010	7 units	PYR	30.8	0.41
				PYR	26.2	0.12
				PYR	31.6	0.01
				PYR	30.4	0.36
				PYR	23.1	0.46
INT				40.8	0.01	
INT				41.5	0.06	
CP-55,940 & Chr2+	m406	m406_230201	3 units	PYR	27.1	1.44
				INT	29.9	0.18
				INT	31.4	0.11
	m432	m432_231019	MUA only			
	m433	m433_231013	7 units	PYR	21.8	0.18
				PYR	25.9	0.18
				PYR	17.5	0.31
				PYR	21.5	0.99
				INT	22.4	0.12
				INT	36.1	0.05
				INT	42.4	0.25
	m435	m435_231016	MUA only			

Supplementary Table 1: Sorted units and their cluster qualities over all acute sessions.

- *What is the clustering quality over many cells? what is the distribution of single unit clustering metrics?*

The clustering quality is sufficient to detect single units. We report a good range of values in isolation distance and ISI violation. Due to the geometry of the probe, the L-Ratio is not a good metric to evaluate data from these devices because the electrodes are not densely packed.

We have included the above table in our supplementary and added the following text in the results section.

Though multi-unit activity (MUA) and power in the spike frequency band is observed in all recordings, we used Kilosort and manual curation using Phy to detect sortable units in 13 out of 19 acute sessions (n=15 animals). Detected units ranged from 1-7 units (median = 3) with 57.9% of units identified as putative pyramidal cells, and 42.1% as putative interneurons. Validated spike sorting metrics⁷⁴ of Isolation Distance ($\mu=26.9$, $\sigma=9.7$), and percentage of spikes that violate the inter-spike interval ($<2\text{ms}$, $\mu=0.53\%$, $\sigma=0.61\%$) are reported.

74. Petersen, P.C., Siegle, J.H., Steinmetz, N.A., Mahallati, S. & Buzsáki, G. CellExplorer: A framework for visualizing and characterizing single neurons. *Neuron* **109**, 3594-3608.e3592 (2021).

- *How reliable are each of these key features (light power, drug area covered, single unit yield, cluster quality) replicated over many experiments?*

We thank the reviewer for this comment. We find our probe to be reliable: across many sessions, we find putative units with sufficient cluster quality (Supplementary table 1), we induced reliable optogenetic manipulation at the unit (fig. 5) and oscillation (fig. 8) level, reliably saw response to CP-55,940 in all drug infusion experiments (fig. 6,7,8), and found that cells were displaced and returned in all infusion experiments (fig. 5).

Following the reviewer's helpful questions, we completed additional experiments and analysis to validate the reliability of our probes. We divided fig. 4 into two separate figures to address the optogenetic manipulation at the unit and oscillation level and added the following text in the results section.

By regulating optical stimulation power, we were able to modulate individual unit spike rate.

a. Example trace of wideband extracellular response to optical stimulation. **b.** The firing rate of sorted cells during optical stimulation and the normalized firing rate across all sessions.

- *This device should be compared to the current gold standards in the field. For all of the above questions, how does this device compare to current standard devices. For example, how does the single unit yield compare to current silicone probes and tetrodes? Furthermore, how does the diameter of this device compare to current silicone probes, tetrodes, optical fibers, and infusion devices. How does the impedance of these electrodes compare to tetrodes and silicone probed?*

We thank the reviewer for the comment. The quality of the local field potential and multiunit activity compares well to the standard silicon probes and tetrodes. The impedance required to detect multiunit recording is $<2\text{M}\Omega$ at 1kHz (Neto et al., *Frontiers in Neuroscience*. 2018) and some silicon probes and most tetrodes have an impedance $<1\text{M}\Omega$. Our devices exhibit $\sim 200\text{ k}\Omega$ at 1kHz (fig. 3d). In terms of single unit yield (via standard spike sorting) our probe is intermediate between single wires and tetrodes (Supplementary Table 1). On top of the recording capability, our probe enables optical stimulation for optogenetics and focal drug infusion for pharmacological intervention. The diameter of our probe is $\sim 150\text{ }\mu\text{m}$ and is in between optical fiber (for optogenetics) and silicon probes. Our device is significantly smaller than a stainless steel cannula which is commonly used to infuse drug into the brain. In addition, we calculated the material and equipment cost of our device to be $< \$10/\text{unit}$, which is significantly lower than silicon probes which are fabricated using clean room facilities.

One future direction of our T-DOpE probes is to densely pack the electrodes to increase the spike sorting capability, which is made possible through the thermal tapering method.

We have added the following discussion in the main text:

On top of the recording capability, our probe enables optical stimulation for optogenetics and focal drug infusion for pharmacological intervention. The diameter of T-DOpE probe is ~150 μm and is in between optical fiber (for optogenetics) and silicon probes. Our novel device is significantly smaller than stainless steel cannula commonly used in intracranial drug infusion.

• *What is the size of each channel, waveguide, and electrode in Fig 2D?*

We thank the reviewer for this question on the size of each feature. We added the following sentence in the main text to clarify the feature size.

The size of the electrodes, microfluidic channels, and optical waveguides are ~25 μm , 30-50 μm , and 30 μm respectively.

• *What is the effect of the optical stimulation on single neurons? The authors should confirm that it is possible to precisely control spike timing of single units, not just alter LFP, as optogenetic manipulation of single units is the common standard in the field.*

We thank the reviewer for the comment on optical stimulation. We agree that addressing optogenetic manipulation of single units is necessary to exemplify the capability of our device. Following the reviewer's suggestions, we conducted additional experiments and analysis. We demonstrate our probe is capable of driving changes in spiking activity along with the changes in LFP. We have added a representative extracellular trace and peristimulus time histograms to address this concern. We show that at low power, we don't excite cells, while at higher power, the firing rate of many cells increase.

We added the following sentence in the results section and modified fig. 5.

By regulating optical stimulation power, we were able to modulate individual unit spike rate.

a. Example trace of wideband extracellular response to optical stimulation. **b.** The firing rate of sorted cells during optical stimulation and the normalized firing rate across all sessions.

• As written, the manuscript dedicates a lot of space to scientific experiments (three figures) but does not provide strong rationale for why these experiments in particular are important. In many ways these findings replicate and confirm prior work that has shown systemic manipulation of cannabinoids affects theta and SWRs or local infusion in anesthetized animals affects SWRs. While this study manipulates cannabinoids locally in awake behaving animals, it is not clear why that manipulation is important or how these results change the way we think about the role of cannabinoids affect hippocampal activity. In principle, I agree it is important to differentiate between local or non-local mechanisms, but I do not understand the significance of local vs non-local cannabinoid actions in hippocampus. I left thinking, what have I learned? The authors could resolve this in different ways. One option would be to minimize the scientific findings and explain this is simply a demonstration of the technology with the aim of replicating prior work with better localization in awake animals. Another option would be to elaborate more on the scientific background, rationale, and hypotheses that guide these experiments. Why do we care about cannabinoids effects on hippocampal activity? Why do we care if the effect is local or not? What are the open questions in this area and why are they important? How do these findings change our mental model of how hippocampus functions?

We thank the reviewer for the perspective on our scientific rationale. We agree we did not spend sufficient text dedicated to conveying the significance of our findings. To address this, we made additions to our main and discussion sections to elaborate and contextualize our experiments and findings within the greater understanding of cannabinoids, learning and memory, and hippocampal oscillations. We specifically identify holes in the literature relevant to, and explained the significance of focal versus systemic infusion.

We refined the introduction:

The hippocampus plays a major role in memory including for learned spatial locations.^{31, 32} During exploration, hippocampal area CA1 (CA1) pyramidal cell (PYR) activity is organized at the theta timescale (6-11 Hz);^{33, 34} during consummatory behaviors and non-rapid eye movement sleep, large depolarization events drive the generation of sharp wave-ripples (SPW-R, 100-250 Hz). Coordinated activity of PYR organized first by theta followed by SPW-R associated replay is thought to support memory consolidation.³⁵ Further, SPW-Rs are an important electrophysiological marker of learning and memory^{36, 37} and causal roles for SPW-Rs in driving specific behaviors have been demonstrated. Behavioral performance is improved when SPW-Rs are extended in duration;³⁸ while disruption/truncation of SPW-Rs in both wake and non-rem sleep states decreases performance.³⁹⁻⁴² In rodents, cannabinoid type-1 receptor (CB1-R) activation leads to neuronal populations losing their temporally structured co-activity, thought to cause the disruption of hippocampal synchronous activity including epileptic seizures (high frequency oscillations),⁴³⁻⁴⁶ as well as theta oscillations and SPW-Rs.^{47, 48} The cannabinoid induced disruption of theta and SPW-Rs is suggested to be a mechanism behind cannabinoid-associated memory impairment in rodents^{47, 49-52} and humans.⁵³⁻⁵⁵ While a mechanism behind seizure disruption has been identified,⁵⁶ the relationship between the CB1R and memory-supporting rhythms remains unclear. It is believed cannabinoids act through CB1R to impair memory by changing the spiking activity of neurons, either through acting directly on cells expressing CB1R in CA1, or indirectly by acting on presynaptic partners.^{52, 57-62} Thus, the precise mechanisms by which CB1R agonists impair hippocampal rhythms have yet to be identified.

Theta and SPW-R oscillations require independent mechanisms of generation,^{33-35, 37, 63} and CB1R expression widely varies across cell types and brain areas,^{51, 58-60, 64, 65} further complicating our understanding of specific cellular/synaptic loci at which cannabinoid signaling occurs. Among CA1 neurons, CB1 receptors are expressed on the axon terminals of by CCK+ basket cells, which act to suppress neurotransmitter release,^{58, 59, 66} and on pyramidal cell dendrites, decreasing excitability.⁶⁷ The theta oscillation in CA1 is strongly supported by inputs from the medial septum,^{33, 63} where CB1Rs are widely expressed including on axon terminals projecting to CA1.⁶⁴ Systemic infusion of CB1 agonists disrupts the theta oscillation and organization of spike timing,^{47, 48} but because this method activates CB1Rs throughout the body and brain it is unable to identify mechanistic loci. Thus, it is currently unclear if the activation of CB1Rs within the medial septum, within CA1, or on projections from the medial septum are responsible for theta disruption. The SPW-R oscillation is generated by the reciprocal interaction of excitatory pyramidal cells and local inhibitory interneurons in CA1,⁶⁸⁻⁷¹ initiated by strong excitatory input from Schaffer collaterals originating in CA3,^{35, 37} whose terminals express CB1 receptors.⁶⁰⁻⁶² SPW-Rs can also be generated experimentally by depolarizing CA1 excitatory neurons.⁶⁸ The cellular-synaptic mechanisms by which cannabinoids disrupt SPW-Rs have remained enigmatic in large part due to the challenging nature of monitoring and manipulating neuronal activity with pharmacological interventions in microcircuits of behaving animals, which our novel probes robustly support. Specifically, our devices are able to deliver the synthetic cannabinoid CP-55,940, a validated tool to study the effect of CB1R activation on neuronal activity,^{47-49, 52} to a local circuit which we monitor with electrophysiology and manipulate further with optogenetics.

We use our novel device to test the hypothesis that activation of CB1 receptors in CA1 weakens the theta oscillation during exploration. Additionally, we tested if activating CB1 receptors expressed by CA1 neurons is sufficient to disrupt SPW-Rs.

We found that regardless of CA3 inputs, CB1R agonism disrupts SPW-R generation, emphasizing the importance of cannabinoid signaling for local circuit computation.

31. Redish, A.D. Beyond the cognitive map: from place cells to episodic memory. (MIT press, 1999).
32. Buzsáki, G. & Moser, E.I. Memory, navigation and theta rhythm in the hippocampal-entorhinal system. *Nature Neuroscience* **16**, 130-138 (2013).
33. Buzsáki, G. Theta Oscillations in the Hippocampus. *Neuron* **33**, 325-340 (2002).
34. Buzsáki, G. Theta rhythm of navigation: Link between path integration and landmark navigation, episodic and semantic memory. *Hippocampus* **15**, 827-840 (2005).
35. Buzsáki, G. Two-stage model of memory trace formation: A role for noisy brain states. *Neuroscience* **31**, 551-570 (1989).
36. Wilson, M.A. & McNaughton, B.L. Reactivation of Hippocampal Ensemble Memories During Sleep. *Science* **265**, 676-679 (1994).
37. Buzsáki, G. Hippocampal sharp wave-ripple: A cognitive biomarker for episodic memory and planning. *Hippocampus* **25**, 1073-1188 (2015).
38. Fernández-Ruiz, A. et al. Long-duration hippocampal sharp wave ripples improve memory. *Science* **364**, 1082-1086 (2019).
39. Girardeau, G., Benchenane, K., Wiener, S.I., Buzsáki, G. & Zugaro, M.B. Selective suppression of hippocampal ripples impairs spatial memory. *Nature Neuroscience* **12**, 1222-1223 (2009).
40. Gridchyn, I., Schoenenberger, P., O'Neill, J. & Csicsvari, J. Assembly-Specific Disruption of Hippocampal Replay Leads to Selective Memory Deficit. *Neuron* **106**, 291-300.e296 (2020).
41. Ego-Stengel, V. & Wilson, M.A. Disruption of ripple-associated hippocampal activity during rest impairs spatial learning in the rat. *Hippocampus* **20**, 1-10 (2010).
42. Jadhav, S.P., Kemere, C., German, P.W. & Frank, L.M. Awake Hippocampal Sharp-Wave Ripples Support Spatial Memory. *Science* **336**, 1454-1458 (2012).
43. Soltesz, I. et al. Weeding out bad waves: towards selective cannabinoid circuit control in epilepsy. *Nature Reviews Neuroscience* **16**, 264-277 (2015).
44. Chesher, G.B., Jackson, D.M. & Malor, R.M. Interaction of $\Delta 9$ -tetrahydrocannabinol and cannabidiol with phenobarbitone in protecting mice from electrically induced convulsions. *Journal of Pharmacy and Pharmacology* **27**, 608-609 (1975).
45. Karler, R. & Turkanis, S.A. SUBACUTE CANNABINOID TREATMENT: ANTICONVULSANT ACTIVITY AND WITHDRAWAL EXCITABILITY IN MICE. *British Journal of Pharmacology* **68**, 479-484 (1980).
46. Wallace, M.J., Wiley, J.L., Martin, B.R. & DeLorenzo, R.J. Assessment of the role of CB1 receptors in cannabinoid anticonvulsant effects. *European Journal of Pharmacology* **428**, 51-57 (2001).
47. Robbe, D. et al. Cannabinoids reveal importance of spike timing coordination in hippocampal function. *Nature Neuroscience* **9**, 1526-1533 (2006).
48. Robbe, D. & Buzsáki, G. Alteration of Theta Timescale Dynamics of Hippocampal Place Cells by a Cannabinoid Is Associated with Memory Impairment. *The Journal of Neuroscience* **29**, 12597 (2009).
49. Lichtman, A.H., Dimen, K.R. & Martin, B.R. Systemic or intrahippocampal cannabinoid administration impairs spatial memory in rats. *Psychopharmacology* **119**, 282-290 (1995).
50. Lichtman, A.H. & Martin, B.R. $\Delta 9$ -Tetrahydrocannabinol impairs spatial memory through a cannabinoid receptor mechanism. *Psychopharmacology* **126**, 125-131 (1996).
51. Abush, H. & Akirav, I. Cannabinoids modulate hippocampal memory and plasticity. *Hippocampus* **20**, 1126-1138 (2010).
52. Wise, L.E., Thorpe, A.J. & Lichtman, A.H. Hippocampal CB1 Receptors Mediate the Memory Impairing Effects of $\Delta 9$ -Tetrahydrocannabinol. *Neuropsychopharmacology* **34**, 2072-2080 (2009).
53. Ranganathan, M. & D'Souza, D.C. The acute effects of cannabinoids on memory in humans: a review. *Psychopharmacology* **188**, 425-444 (2006).
54. Ilan, A.B., Smith, M.E. & Gevins, A. Effects of marijuana on neurophysiological signals of working and episodic memory. *Psychopharmacology* **176**, 214-222 (2004).
55. Curran, V.H., Brignell, C., Fletcher, S., Middleton, P. & Henry, J. Cognitive and subjective dose-response effects of acute oral $\Delta 9$ -tetrahydrocannabinol (THC) in infrequent cannabis users. *Psychopharmacology* **164**, 61-70 (2002).
56. Farrell, J.S. et al. In vivo endocannabinoid dynamics at the timescale of physiological and pathological neural activity. *Neuron* **109**, 2398-2403.e2394 (2021).
57. Sullivan, J.M. Mechanisms of Cannabinoid-Receptor-Mediated Inhibition of Synaptic Transmission in Cultured Hippocampal Pyramidal Neurons. *Journal of Neurophysiology* **82**, 1286-1294 (1999).

58. Katona, I. et al. Presynaptically Located CB1 Cannabinoid Receptors Regulate GABA Release from Axon Terminals of Specific Hippocampal Interneurons. *The Journal of Neuroscience* **19**, 4544 (1999).
59. Losonczy, A., Biró, A.A. & Nusser, Z. Persistently active cannabinoid receptors mute a subpopulation of hippocampal interneurons. *Proceedings of the National Academy of Sciences* **101**, 1362-1367 (2004).
60. Takahashi, K.A. & Castillo, P.E. The CB1 cannabinoid receptor mediates glutamatergic synaptic suppression in the hippocampus. *Neuroscience* **139**, 795-802 (2006).
61. Sandler, R.A., Fetterhoff, D., Hampson, R.E., Deadwyler, S.A. & Marmarelis, V.Z. Cannabinoids disrupt memory encoding by functionally isolating hippocampal CA1 from CA3. *PLOS Computational Biology* **13**, e1005624 (2017).
62. Maier, N. et al. Cannabinoids disrupt hippocampal sharp wave-ripples via inhibition of glutamate release. *Hippocampus* **22**, 1350-1362 (2012).
63. Vandecasteele, M. et al. Optogenetic activation of septal cholinergic neurons suppresses sharp wave ripples and enhances theta oscillations in the hippocampus. *Proceedings of the National Academy of Sciences* **111**, 13535-13540 (2014).
64. Nyíri, G. et al. GABAB and CB1 cannabinoid receptor expression identifies two types of septal cholinergic neurons. *European Journal of Neuroscience* **21**, 3034-3042 (2005).
65. Busquets-Garcia, A., Bains, J. & Marsicano, G. CB1 Receptor Signaling in the Brain: Extracting Specificity from Ubiquity. *Neuropsychopharmacology* **43**, 4-20 (2018).
66. Neu, A., Földy, C. & Soltesz, I. Postsynaptic origin of CB1-dependent tonic inhibition of GABA release at cholecystokinin-positive basket cell to pyramidal cell synapses in the CA1 region of the rat hippocampus. *The Journal of Physiology* **578**, 233-247 (2007).
67. Maroso, M. et al. Cannabinoid Control of Learning and Memory through HCN Channels. *Neuron* **89**, 1059-1073 (2016).
68. Stark, E. et al. Pyramidal Cell-Interneuron Interactions Underlie Hippocampal Ripple Oscillations. *Neuron* **83**, 467-480 (2014).
69. Klausberger, T. et al. Brain-state- and cell-type-specific firing of hippocampal interneurons in vivo. *Nature* **421**, 844-848 (2003).
70. Royer, S. et al. Control of timing, rate and bursts of hippocampal place cells by dendritic and somatic inhibition. *Nature Neuroscience* **15**, 769-775 (2012).
71. Dániel, S., Szabolcs, K., Tamás, F.F., Norbert, H. & Attila, I.G. Mechanisms of Sharp Wave Initiation and Ripple Generation. *The Journal of Neuroscience* **34**, 11385 (2014).

We refined the discussion:

Hippocampal circuit activity is critical for episodic and spatial memory. Hippocampal theta (~6-10 Hz), gamma (~35-80 Hz), and sharp wave-ripple (SPW-R, ~100-250 Hz) oscillations all contribute to mnemonic functions of the circuitry.^{33-37, 73} Cannabinoids impair memory and alter CA1 rhythms, suggesting a possible role for local CB1R in hippocampal activity and oscillations supporting memory.^{47, 48} However, CB1Rs are expressed in multiple hippocampal sub-regions (e.g., both CA1 and CA3) and cell types (e.g., excitatory pyramidal cells and inhibitory interneurons),^{51, 58-60, 64, 66} and thus the specific loci of action of cannabinoids on CA1 activity remains unknown. The development of our novel device has enabled us to further our understanding of the interaction of CB1R and hippocampal oscillations specifically through the focal infusion of agonist during behavior and optogenetic initiation of SPW-Rs.

Administration of CB1R agonists disrupts theta while keeping neuronal firing rate intact, through scrambling temporal coding by cell assemblies.^{47, 48} Thus while it is clear that cannabinoids disrupt neural circuits underlying theta, these studies were achieved through systemic administration of cannabinoids in rats. Until our study, it was unknown if cannabinoids were disrupting theta generated upstream of CA1, or within local circuitry. The medial septum, a proposed theta generator, has glutamatergic, cholinergic, and GABAergic connections to CA1.⁶⁴ These GABAergic and cholinergic neurons express CB1 receptors, likely inhibiting vesicle release at axon terminals.^{57, 58} One hypothesis is that systemic administration of cannabinoids may decrease theta by disrupting circuitry within the medial septum, which prevents effective theta drive in CA1. An alternative

hypothesis is that cannabinoids bind to CB1 receptors expressed in CA1, either on CA1 neurons or terminals of inputs into CA1, altering precise timing of neurotransmitter release resulting in disrupted oscillations.⁴⁸ We demonstrated with our T-DOpE probe that local CA1 infusion of CP-55,940 was sufficient to disrupt theta (**fig. 6e**), while maintaining the relationship between velocity and theta power (**Supplementary Table 2**). This is the first demonstration in behaving animals that focal CB1R activation in CA1 is sufficient to weaken the theta oscillation.

Cannabinoids inhibit SPW-Rs in rats,⁴⁷ suggesting a potential mechanism underlying the memory impairment associated with cannabinoid use in humans. This effect was attributed to reduced glutamate release in CA1 from CA3 afferents,^{47, 61, 62} although CB1Rs are expressed in many locations within CA1, leaving the specific mechanism unknown. Importantly, activation of CB1Rs on medial septal inputs would be expected to increase not decrease SPW-R rate (activation of the medial septum decreases SPW-R rate).⁶³ Both CA3 and CA1 neurons express CB1Rs.^{51, 58-60, 64} CB1Rs on CA3 axon terminals of the Schaffer collaterals have been shown to act to suppress glutamate release.^{51, 60} CA1 PYR express CB1Rs localized on their dendrites, activation has been shown to decrease post-synaptic excitability by increasing membrane conductances.⁶⁷ CB1Rs are highly enriched in CA1 INT where they inhibit GABA release.^{58, 59} Given the ambiguous relationship between locus of CB1R expression and SPW-R generation, it was unclear if cannabinoids were acting on excitatory drive from CA3 or local circuit interactions within CA1. Our novel device allows us to directly interrogate CA1 through local drug delivery and optogenetic stimulation. Our experiments show that CB1R activation is sufficient to disrupt SPW-Rs independent of CA3 synaptic transmission (**fig. 8e**), even if the network is sufficiently activated by optogenetic depolarization of CA1 PYR (activating ChR2 expressing PYRs is sufficient to generate CA1 ripples).⁶⁸ (**fig. 8d**). Thus while previous studies suggested that CB1R inhibit SPW-Rs by cutting off communication from other areas,^{61, 62} our findings demonstrate the importance of physiological cannabinoid activity in the CA1 local circuit. Together, these findings reveal novel mechanisms by which cannabinoids disrupt specific hippocampal rhythms and suggest dysregulation of this tightly controlled system within the CA1 circuit may directly lead to memory impairment.

33. Buzsáki, G. Theta Oscillations in the Hippocampus. *Neuron* **33**, 325-340 (2002).

34. Buzsáki, G. Theta rhythm of navigation: Link between path integration and landmark navigation, episodic and semantic memory. *Hippocampus* **15**, 827-840 (2005).

35. Buzsáki, G. Two-stage model of memory trace formation: A role for noisy brain states. *Neuroscience* **31**, 551-570 (1989).

36. Wilson, M.A. & McNaughton, B.L. Reactivation of Hippocampal Ensemble Memories During Sleep. *Science* **265**, 676-679 (1994).

37. Buzsáki, G. Hippocampal sharp wave-ripple: A cognitive biomarker for episodic memory and planning. *Hippocampus* **25**, 1073-1188 (2015).

47. Robbe, D. et al. Cannabinoids reveal importance of spike timing coordination in hippocampal function. *Nature Neuroscience* **9**, 1526-1533 (2006).

48. Robbe, D. & Buzsáki, G. Alteration of Theta Timescale Dynamics of Hippocampal Place Cells by a Cannabinoid Is Associated with Memory Impairment. *The Journal of Neuroscience* **29**, 12597 (2009).

51. Abush, H. & Akirav, I. Cannabinoids modulate hippocampal memory and plasticity. *Hippocampus* **20**, 1126-1138 (2010).

57. Sullivan, J.M. Mechanisms of Cannabinoid-Receptor-Mediated Inhibition of Synaptic Transmission in Cultured Hippocampal Pyramidal Neurons. *Journal of Neurophysiology* **82**, 1286-1294 (1999).

58. Katona, I. et al. Presynaptically Located CB1 Cannabinoid Receptors Regulate GABA Release from Axon Terminals of Specific Hippocampal Interneurons. *The Journal of Neuroscience* **19**, 4544 (1999).
59. Losonczy, A., Biró, A.A. & Nusser, Z. Persistently active cannabinoid receptors mute a subpopulation of hippocampal interneurons. *Proceedings of the National Academy of Sciences* **101**, 1362-1367 (2004).
60. Takahashi, K.A. & Castillo, P.E. The CB1 cannabinoid receptor mediates glutamatergic synaptic suppression in the hippocampus. *Neuroscience* **139**, 795-802 (2006).
61. Sandler, R.A., Fetterhoff, D., Hampson, R.E., Deadwyler, S.A. & Marmarelis, V.Z. Cannabinoids disrupt memory encoding by functionally isolating hippocampal CA1 from CA3. *PLOS Computational Biology* **13**, e1005624 (2017).
62. Maier, N. et al. Cannabinoids disrupt hippocampal sharp wave-ripples via inhibition of glutamate release. *Hippocampus* **22**, 1350-1362 (2012).
63. Vandecasteele, M. et al. Optogenetic activation of septal cholinergic neurons suppresses sharp wave ripples and enhances theta oscillations in the hippocampus. *Proceedings of the National Academy of Sciences* **111**, 13535-13540 (2014).
64. Nyíri, G. et al. GABAB and CB1 cannabinoid receptor expression identifies two types of septal cholinergic neurons. *European Journal of Neuroscience* **21**, 3034-3042 (2005).
66. Neu, A., Földy, C. & Soltesz, I. Postsynaptic origin of CB1-dependent tonic inhibition of GABA release at cholecystinin-positive basket cell to pyramidal cell synapses in the CA1 region of the rat hippocampus. *The Journal of Physiology* **578**, 233-247 (2007).
67. Maroso, M. et al. Cannabinoid Control of Learning and Memory through HCN Channels. *Neuron* **89**, 1059-1073 (2016).
68. Stark, E. et al. Pyramidal Cell-Interneuron Interactions Underlie Hippocampal Ripple Oscillations. *Neuron* **83**, 467-480 (2014).
73. Buzsáki, G., Lai-Wo S, L. & Vanderwolf, C.H. Cellular bases of hippocampal EEG in the behaving rat. *Brain Research Reviews* **6**, 139-171 (1983).

• *Several figures have n of 2,3, or 4 animals. This is too few to draw conclusions. Studies should be replicated in at least 5 animals.*

We thank the reviewer for the comment regarding the power of our findings. We have increased the animal number in all animal experiments. To validate the physical cell displacement of focal infusion, we increased the number of animals for saline infusion to 5. We also increased the number of animals in our experiments and controls regarding the effect of CP-55,940. We report data from an n=5 animals for our CP-55,940 and vehicle control experiments. We increased the number of animals in our optogenetic experiments to 4, which was the most we could accomplish in the time allotted.

We have revised the figures (fig. 5,6,7,8), added supplementary table 1,2 and the corresponding text to reflect the additional experiments.

• *Why is the relationship between speed and theta power different between the baseline conditions in Fig 5f and 5g? These are similar experiments and the relationship should be similar.*

We thank the reviewer for this comment. Across animals and sessions, different mice exhibit different running behavior in terms of time spent running, average velocity, and number of and length of running epochs. Importantly, however, the effect of drug on the relationship between running and theta power is consistent across animals. (Supplementary fig. 7 and supplementary table 2).

We have added the following sentence into the discussion section to clarify the relationship between speed and theta power.

Across animals and sessions, different mice exhibit different running behavior in terms of time spent running, average velocity, and number of and length of running epochs. Importantly, however, the effect of drug on the relationship between running and theta power is consistent across animals (**Supplementary fig. 8 and supplementary table 2**).

- *The statistical approach, primarily student's T-tests, ignores interdependence in data recorded from the same animal. When data recorded from the same animal is treated as independent statistical power is inflated. A more appropriate approach would be linear mixed models. Furthermore, the authors should not assume data is normally distributed without testing whether that is the case.*

We thank the reviewer for raising this concern and we have performed new analyses. Specifically, we conducted paired and/or non-parametric tests when appropriate. We used the Shapiro-Wilk normality test to check for normality and used either paired T-test or Wilcoxon signed rank test to observe statistical significance. We used a linear mixed model to determine the relationship between theta power and velocity (supplementary fig. 8 and supplementary table 2).

We have added the following sentences into the method and result section and modified the figures (fig. 6,7,8) accordingly:

The two comparison groups were tested with Shapiro-Wilk normality test to check if the groups are normally distributed. If the two groups were normally distributed, we used the paired T-test. For those that are not normally distributed, the experimental designs with two comparison groups were analyzed by Wilcoxon signed rank tests. Linear mixed-effects model was used to characterize the relation between the running speed and the theta power.

In **supplementary table 2**, we used linear mixed-estimate model to fit the theta power as a function of an intercept and running speed grouped by infusion condition. The average speed and standard deviation of all 10 sessions suggest that the mice's running behavior was not influenced by drug infusion.

Minor concerns

- *The text on the panels in Figure 2d is really hard to read*

We thank the reviewer for this comment. We have changed the font size of the text in Fig. 2d.

- *The saline and vehicle infusion controls are very important and should be in the main figures*

We thank the reviewer for the comment. We agree that the saline and vehicle infusion controls are very important. Therefore, we conducted additional infusion experiments to increase sample size (n=5) and included the results in the main text.

To validate that the transient silencing of cells is a result of the physical cell displacement^{72, 78, 79} instead of pharmacological influence, we performed a detailed comparison of the dynamic spike recording during the infusion of saline, vehicle, and drug with vehicle. We observed transient absences of spikes after infusion of either saline, vehicle, or drug+vehicle (**fig. 5d, supplementary fig. 7**).

72. Korposh, S., James, S.W., Lee, S.-W. & Tatam, R.P. Tapered Optical Fibre Sensors: Current Trends and Future Perspectives *Sensors* **19**, 2294 (2019).

78. Yoon, Y. et al. Neural probe system for behavioral neuropharmacology by bi-directional wireless drug delivery and electrophysiology in socially interacting mice. *Nature Communications* **13**, 5521 (2022).

79. Shin, H. et al. Neural probes with multi-drug delivery capability. *Lab on a Chip* **15**, 3730-3737 (2015).

d. Normalized firing rate of sorted cells in saline, vehicle, and drug+vehicle infusion. Note shortly after the infusion, the firing rate of all cells drastically decrease not due to pharmacological influence but to the physical displacement of the cells from the recording sites.

- Were experimenters blind to manipulation conditions during experiments and analysis?

We thank the reviewer for the comment. Due to the nature of the experiment where the researcher had to prepare the drug immediately before the experiment, they were not blinded to the manipulation conditions. However, analysis pipelines including ripple detection and theta power detection were consistent across animals and were not manipulated for experimental condition.

Reviewer #3 (Remarks to the Author):

The authors devised a method to fabricate flexible, multifunctional probes (T-DOpE) with electrodes, waveguides, and microfluidic channels in one device via a thermal drawing process. The diameter of the probe tapers down from 2 mm, allowing for easy backend connections, down to 150 μm which enables high spatial specificity. T-DOpE was used to investigate the mechanism of action of CP-55,940, a cannabinoid, in the CA1 region in the hippocampus. First, the authors showed that focal CA1 cannabinoid receptor (CB1R) agonism, not systemic agonism, is sufficient to reduce theta oscillations in CA1 and to reduce short-wave ripple (SPW-R) rates. Using mice expressing channel rhodopsin (ChR2), which optogenetically induces SPW-R, in CA1, the authors addressed the question of whether the reason SPW-R rate decreases after CP-55,940 infusion is due to

CB1R agonism at the synaptic terminals of CA3 inputs and/or hippocampal interneurons. SPW-R was locally induced in CA1 via optical stimulation of ChR2. It was found that SPW-R decreased as a result of CP-55,940 infusion, confirming that agonism of CB1R by CP-55,940 in CA1 interneurons is responsible for SPW-R suppression. The multifunctional probe and its use for a specific neuroscience study is certainly very interesting. With that being said, the proposed device seems to be a relatively trivial step on top of the thermal drawing process that has been reported previously to fabricate multifunctional probes (Canales et al. Nat Biotechnol, 2015). So it is not clear from the paper where is the innovation of this method. Also, not much characterization is done on the device either. Furthermore, since the neuroscience conducted in this study is very niche, the broader implications of the findings are not clear. Below are some specific questions and suggestions.

We thank the reviewer for the feedback. Multifunctional fiber probe was first demonstrated in Canales et al. Nat Biotechnol, 2015. However, the application of the probe in the neuroscience field has been limited due to the intrinsic challenges of the fabrication method and connectorization. Here, we present a drastically different approach merging the two cutting edge technologies, i.e., thermal fiber drawing and multi-material tapering. This merge opens new possibilities of broad distribution of T-DOPe probes in the neuroscience community and a wide selection of new materials, geometrical arrangement, and functionalities that can be embedded in this technology.

More specifically, previous thermally drawn fibers only allow one individually addressable optical waveguide, a low number of electrodes (≤ 6) and microfluidic channels (≤ 2) to be connected per fiber. Here, using our novel approach, the connectorization is no longer a limiting factor in our probe design. The tapering of silica has enabled significant advancement in the neuroscience field such as glass micropipettes used in patch clamp recording. And more recently, tapered silica fiber was developed for multisite photometry (Pisano et al., Nature Methods. 2019) and single neuron recording (LeChasseur et al., Nature Methods. 2011). In our work, we demonstrated for the first time, multi-material tapering for multifunctional neural probes.

Previous studies primarily assigned the cannabinoid induced disorganized spike timing to functional disconnect between CA1 and other brain areas. Our results demonstrate that cannabinoid modulation of local circuit interactions is likely to be as important as inter-area interactions. These findings thus inform any and all future investigations of the effects of cannabinoids on the brain and behavior. Additionally, due to the recent exponential rise in cannabis use by US population, the study of its neural mechanisms is transitioning from a niche topic to a mainstream one.

We have revised the introduction and discussion sections accordingly to highlight the innovation of this work. We have also performed additional experiments to characterize the device as detailed below. We hope the reviewer finds that these additional results and discussion significantly strengthen our manuscript.

Questions:

1. What are the optical properties of the probe like? Transmission spectra would be useful.

We thank the reviewer for the comment on the optical properties. We have conducted additional experiments and measured the optical transmission spectrum using a DH-2000 light source (Ocean Insight) and FLAME-S-XR1 spectrometer (Ocean Insight). We have added a transmission spectrum plot (400-750nm) to our supplementary information and added the following sentence in the results section:

The transmission spectrum of the optical waveguide is included in the supplementary fig. 4.

Supplementary Figure 4: Transmission Spectrum of the optical waveguide

2. What are the mechanical properties of the probe like? What is the stiffness? What is the ultimate flexural strength?

We thank the reviewer for comments on adding more mechanical properties of the probe. Following the reviewer's suggestions, we used a dynamic mechanical analyzer (DMA Q800 from TA instrument) to measure the bending stiffness of the three designs and a stainless steel wire, which has the same diameter as our probes. The bending stiffness of the probe was smaller than the stiffness of the wire. The ultimate flexural strength of polycarbonate is ~60 MPa. We have added a video of the probe bending to 90° with a radius of curvature of 1.07 cm without breaking to demonstrate the flexibility of the probe (supplementary video 1). In the results section, we measured the impedance of the electrodes before and after bending (supplementary video 1) to demonstrate this bending does not affect the functionality of the probe. We have added the stiffness plot into the supplementary and added the following sentence:

Using a dynamic mechanical analyzer, we measured the bending stiffness of the three designs and stainless steel wire, included in **supplementary fig. 5**. Note that all designs have a much lower stiffness than the stainless steel wire with the same diameter.

Supplementary Figure 5: Stiffness vs Frequency measurement using a dynamic mechanical analyzer

3. Depending on the mechanical properties, the probe could break if handled in an incorrect manner. Is this a concern?

We thank the reviewer for the question of the fragility of the probe. The main material of the probe is polycarbonate. The elongation to fracture of polycarbonate is 110%, which means, unlike some polymers that shatter like glass, such as acrylic, polycarbonate can undergo large deformation without breaking or cracking. As shown in the supplementary video 1, the probe can be deformed and still be functional. The user should be able to drop the probe, and still use it. However, stepping on the probe might collapse the microfluidic channels of the probe.

We included the following sentence in the results section to clarify the robustness of the probes.

The elongation to fracture of polycarbonate is 110%, which means, unlike some polymers that shatter like glass, such as acrylic, polycarbonate can undergo large deformation without breaking or cracking.

4. For Figures 5f and 5g where the theta power is plotted against the speed, is the difference between the two linear regressions statistically significant? A two-way ANOVA or a statistics test of the author's choice would be helpful. Also, what is the correlation strength for the linear regressions? Depending on these quantities, the data presented in Figure 5h could be misleading.

We thank the reviewer for raising this concern and we have performed new analyses. Specifically, we conducted paired and/or non-parametric tests when appropriate. We used the Shapiro-Wilk normality test to check for normality and used either paired T-test or Wilcoxon signed rank test to observe statistical significance. We used a linear mixed model to determine the relationship between theta power and velocity (supplementary fig. 8 and supplementary table 2).

Across animals and sessions, different mice exhibit different running behavior in terms of time spent running, average velocity, and number of and length of running epochs. Importantly, however, the effect of drug on the relationship between running and theta power is consistent across animals. (Supplementary fig. 8 and supplementary table 2).

We have added the following sentences to the discussion and method section and modified the figures (fig. 6,7,8) accordingly:

Across animals and sessions, different mice exhibit different running behavior in terms of time spent running, average velocity, and number of and length of running epochs. Importantly, however, the effect of drug on the relationship between running and theta power is consistent across animals. (Supplementary fig. 8 and supplementary table 2).

The two comparison groups were tested with Shapiro-Wilk normality test to check if the groups are normally distributed. If the two groups were normally distributed, we used the paired T-test. For those that are not normally distributed, the experimental designs with two comparison groups were analyzed by Wilcoxon signed rank tests. Linear mixed-effects model was used to characterize the relation between the running speed and the theta power.

5. What are the broader implications of the findings? How does it help the field advance its understanding of the hippocampus and memory? Do the findings have the potential to lead to practical applications?

We thank for reviewer for their questions. To better contextualize our findings we have modified the introduction:

The hippocampus plays a major role in memory including for learned spatial locations.^{31, 32} During exploration, hippocampal area CA1 (CA1) pyramidal cell (PYR) activity is organized at the theta timescale (6-11 Hz);^{33, 34} during consummatory behaviors and non-rapid eye movement sleep, large depolarization events drive the generation of sharp wave-ripples (SPW-R, 100-250 Hz). Coordinated activity of PYR organized first by theta followed by SPW-R associated replay is thought to support memory consolidation.³⁵ Further, SPW-Rs are an important electrophysiological marker of learning and memory^{36, 37} and causal roles for SPW-Rs in driving specific behaviors have been demonstrated. Behavioral performance is improved when SPW-Rs are extended in duration;³⁸ while disruption/truncation of SPW-Rs in both wake and non-rem sleep states decreases performance.³⁹⁻⁴² In rodents, cannabinoid type-1 receptor (CB1-R) activation leads to neuronal populations losing their temporally structured co-activity, thought to cause the disruption of hippocampal synchronous activity including epileptic seizures (high frequency oscillations),⁴³⁻⁴⁶ as well as theta oscillations and SPW-Rs.^{47, 48} The

cannabinoid induced disruption of theta and SPW-Rs is suggested to be a mechanism behind cannabinoid-associated memory impairment in rodents^{47, 49-52} and humans.⁵³⁻⁵⁵ While a mechanism behind seizure disruption has been identified,⁵⁶ the relationship between the CB1R and memory-supporting rhythms remains unclear. It is believed cannabinoids act through CB1R to impair memory by changing the spiking activity of neurons, either through acting directly on cells expressing CB1R in CA1, or indirectly by acting on presynaptic partners.^{52, 57-62} Thus, the precise mechanisms by which CB1R agonists impair hippocampal rhythms remain unclear.

Theta and SPW-R oscillations require independent mechanisms of generation,^{33-35, 37, 63} and CB1R expression widely varies across cell types and brain areas,^{51, 58-60, 64, 65} further complicating our understanding of specific cellular/synaptic loci at which cannabinoid signaling occurs. Among CA1 neurons, CB1 receptors are expressed on the axon terminals of by CCK+ basket cells, which act to suppress neurotransmitter release,^{58, 59, 66} and on pyramidal cell dendrites, decreasing excitability.⁶⁷ The theta oscillation in CA1 is strongly supported by inputs from the medial septum,^{33, 63} where CB1Rs are widely expressed including on axon terminals projecting to CA1.⁶⁴ Systemic infusion of CB1 agonists disrupts the theta oscillation and organization of spike timing,^{47, 48} but because this method activates CB1Rs throughout the body and brain it is unable to identify mechanistic loci. Thus, it is currently unclear if the activation of CB1Rs within the medial septum, within CA1, or on projections from the medial septum are responsible for theta disruption. The SPW-R oscillation is generated by the reciprocal interaction of excitatory pyramidal cells and local inhibitory interneurons in CA1,⁶⁸⁻⁷¹ initiated by strong excitatory input from Schaffer collaterals originating in CA3,^{35, 37} whose terminals express CB1 receptors.⁶⁰⁻⁶² SPW-Rs can also be generated experimentally by depolarizing CA1 excitatory neurons.⁶⁸ The cellular-synaptic mechanisms by which cannabinoids disrupt SPW-Rs have remained enigmatic in large part due to the challenging nature of monitoring and manipulating neuronal activity with pharmacological interventions in microcircuits of behaving animals, which our novel probes robustly support. Specifically our devices are able to deliver the synthetic cannabinoid CP-55,940, a validated tool to study the effect of CB1R activation on neuronal activity,^{47-49, 52} to a local circuit which we monitor with electrophysiology and manipulate further with optogenetics.

We found that regardless of CA3 inputs, CB1R agonism disrupts SPW-R generation, emphasizing the importance of cannabinoid signaling for local circuit computation.

31. Redish, A.D. Beyond the cognitive map: from place cells to episodic memory. (MIT press, 1999).

32. Buzsáki, G. & Moser, E.I. Memory, navigation and theta rhythm in the hippocampal-entorhinal system. *Nature Neuroscience* **16**, 130-138 (2013).

33. Buzsáki, G. Theta Oscillations in the Hippocampus. *Neuron* **33**, 325-340 (2002).

34. Buzsáki, G. Theta rhythm of navigation: Link between path integration and landmark navigation, episodic and semantic memory. *Hippocampus* **15**, 827-840 (2005).

35. Buzsáki, G. Two-stage model of memory trace formation: A role for noisy brain states. *Neuroscience* **31**, 551-570 (1989).

36. Wilson, M.A. & McNaughton, B.L. Reactivation of Hippocampal Ensemble Memories During Sleep. *Science* **265**, 676-679 (1994).

37. Buzsáki, G. Hippocampal sharp wave-ripple: A cognitive biomarker for episodic memory and planning. *Hippocampus* **25**, 1073-1188 (2015).

38. Fernández-Ruiz, A. et al. Long-duration hippocampal sharp wave ripples improve memory. *Science* **364**, 1082-1086 (2019).
39. Girardeau, G., Benchenane, K., Wiener, S.I., Buzsáki, G. & Zugaro, M.B. Selective suppression of hippocampal ripples impairs spatial memory. *Nature Neuroscience* **12**, 1222-1223 (2009).
40. Gridchyn, I., Schoenenberger, P., O'Neill, J. & Csicsvari, J. Assembly-Specific Disruption of Hippocampal Replay Leads to Selective Memory Deficit. *Neuron* **106**, 291-300.e296 (2020).
41. Ego-Stengel, V. & Wilson, M.A. Disruption of ripple-associated hippocampal activity during rest impairs spatial learning in the rat. *Hippocampus* **20**, 1-10 (2010).
42. Jadhav, S.P., Kemere, C., German, P.W. & Frank, L.M. Awake Hippocampal Sharp-Wave Ripples Support Spatial Memory. *Science* **336**, 1454-1458 (2012).
43. Soltesz, I. et al. Weeding out bad waves: towards selective cannabinoid circuit control in epilepsy. *Nature Reviews Neuroscience* **16**, 264-277 (2015).
44. Chesher, G.B., Jackson, D.M. & Malor, R.M. Interaction of $\Delta 9$ -tetrahydrocannabinol and cannabidiol with phenobarbitone in protecting mice from electrically induced convulsions. *Journal of Pharmacy and Pharmacology* **27**, 608-609 (1975).
45. Karler, R. & Turkanis, S.A. SUBACUTE CANNABINOID TREATMENT: ANTICONVULSANT ACTIVITY AND WITHDRAWAL EXCITABILITY IN MICE. *British Journal of Pharmacology* **68**, 479-484 (1980).
46. Wallace, M.J., Wiley, J.L., Martin, B.R. & DeLorenzo, R.J. Assessment of the role of CB1 receptors in cannabinoid anticonvulsant effects. *European Journal of Pharmacology* **428**, 51-57 (2001).
47. Robbe, D. et al. Cannabinoids reveal importance of spike timing coordination in hippocampal function. *Nature Neuroscience* **9**, 1526-1533 (2006).
48. Robbe, D. & Buzsáki, G. Alteration of Theta Timescale Dynamics of Hippocampal Place Cells by a Cannabinoid Is Associated with Memory Impairment. *The Journal of Neuroscience* **29**, 12597 (2009).
49. Lichtman, A.H., Dimen, K.R. & Martin, B.R. Systemic or intrahippocampal cannabinoid administration impairs spatial memory in rats. *Psychopharmacology* **119**, 282-290 (1995).
50. Lichtman, A.H. & Martin, B.R. $\Delta 9$ -Tetrahydrocannabinol impairs spatial memory through a cannabinoid receptor mechanism. *Psychopharmacology* **126**, 125-131 (1996).
51. Abush, H. & Akirav, I. Cannabinoids modulate hippocampal memory and plasticity. *Hippocampus* **20**, 1126-1138 (2010).
52. Wise, L.E., Thorpe, A.J. & Lichtman, A.H. Hippocampal CB1 Receptors Mediate the Memory Impairing Effects of $\Delta 9$ -Tetrahydrocannabinol. *Neuropsychopharmacology* **34**, 2072-2080 (2009).
53. Ranganathan, M. & D'Souza, D.C. The acute effects of cannabinoids on memory in humans: a review. *Psychopharmacology* **188**, 425-444 (2006).
54. Ilan, A.B., Smith, M.E. & Gevins, A. Effects of marijuana on neurophysiological signals of working and episodic memory. *Psychopharmacology* **176**, 214-222 (2004).
55. Curran, V.H., Brignell, C., Fletcher, S., Middleton, P. & Henry, J. Cognitive and subjective dose-response effects of acute oral $\Delta 9$ -tetrahydrocannabinol (THC) in infrequent cannabis users. *Psychopharmacology* **164**, 61-70 (2002).
56. Farrell, J.S. et al. In vivo endocannabinoid dynamics at the timescale of physiological and pathological neural activity. *Neuron* **109**, 2398-2403.e2394 (2021).
57. Sullivan, J.M. Mechanisms of Cannabinoid-Receptor-Mediated Inhibition of Synaptic Transmission in Cultured Hippocampal Pyramidal Neurons. *Journal of Neurophysiology* **82**, 1286-1294 (1999).
58. Katona, I. et al. Presynaptically Located CB1 Cannabinoid Receptors Regulate GABA Release from Axon Terminals of Specific Hippocampal Interneurons. *The Journal of Neuroscience* **19**, 4544 (1999).
59. Losonczy, A., Biró, Á.A. & Nusser, Z. Persistently active cannabinoid receptors mute a subpopulation of hippocampal interneurons. *Proceedings of the National Academy of Sciences* **101**, 1362-1367 (2004).
60. Takahashi, K.A. & Castillo, P.E. The CB1 cannabinoid receptor mediates glutamatergic synaptic suppression in the hippocampus. *Neuroscience* **139**, 795-802 (2006).
61. Sandler, R.A., Fetterhoff, D., Hampson, R.E., Deadwyler, S.A. & Marmarelis, V.Z. Cannabinoids disrupt memory encoding by functionally isolating hippocampal CA1 from CA3. *PLOS Computational Biology* **13**, e1005624 (2017).
62. Maier, N. et al. Cannabinoids disrupt hippocampal sharp wave-ripples via inhibition of glutamate release. *Hippocampus* **22**, 1350-1362 (2012).
63. Vandecasteele, M. et al. Optogenetic activation of septal cholinergic neurons suppresses sharp wave ripples and enhances theta oscillations in the hippocampus. *Proceedings of the National Academy of Sciences* **111**, 13535-13540 (2014).
64. Nyíri, G. et al. GABAB and CB1 cannabinoid receptor expression identifies two types of septal cholinergic neurons. *European Journal of Neuroscience* **21**, 3034-3042 (2005).
65. Busquets-Garcia, A., Bains, J. & Marsicano, G. CB1 Receptor Signaling in the Brain: Extracting Specificity from Ubiquity. *Neuropsychopharmacology* **43**, 4-20 (2018).
66. Neu, A., Földy, C. & Soltesz, I. Postsynaptic origin of CB1-dependent tonic inhibition of GABA release at cholecystokinin-positive basket cell to pyramidal cell synapses in the CA1 region of the rat hippocampus. *The Journal of Physiology* **578**, 233-247 (2007).

67. Maroso, M. et al. Cannabinoid Control of Learning and Memory through HCN Channels. *Neuron* **89**, 1059-1073 (2016).
68. Stark, E. et al. Pyramidal Cell-Interneuron Interactions Underlie Hippocampal Ripple Oscillations. *Neuron* **83**, 467-480 (2014).
69. Klausberger, T. et al. Brain-state- and cell-type-specific firing of hippocampal interneurons in vivo. *Nature* **421**, 844-848 (2003).
70. Royer, S. et al. Control of timing, rate and bursts of hippocampal place cells by dendritic and somatic inhibition. *Nature Neuroscience* **15**, 769-775 (2012).
71. Dániel, S., Szabolcs, K., Tamás, F.F., Norbert, H. & Attila, I.G. Mechanisms of Sharp Wave Initiation and Ripple Generation. *The Journal of Neuroscience* **34**, 11385 (2014).

We modified the discussion:

Our novel device allows us to directly interrogate CA1 through local drug delivery and optogenetic stimulation. Our experiments show that CB1R activation is sufficient to disrupt SPW-Rs independent of CA3 synaptic transmission (**fig. 8e**), even if the network is sufficiently activated by optogenetic depolarization of CA1 PYR (activating ChR2 expressing PYRs is sufficient to generate CA1 ripples).⁶⁸ (**fig. 8d**). Thus while previous studies suggested that CB1R inhibit SPW-Rs by cutting off communication from other areas,^{61, 62} our findings demonstrate the importance of physiological cannabinoid activity in the CA1 local circuit. Together, these findings reveal novel mechanisms by which cannabinoids disrupt specific hippocampal rhythms and suggest dysregulation of this tightly controlled system within the CA1 circuit may directly lead to memory impairment.

61. Sandler, R.A., Fetterhoff, D., Hampson, R.E., Deadwyler, S.A. & Marmarelis, V.Z. Cannabinoids disrupt memory encoding by functionally isolating hippocampal CA1 from CA3. *PLOS Computational Biology* **13**, e1005624 (2017).

62. Maier, N. et al. Cannabinoids disrupt hippocampal sharp wave-ripples via inhibition of glutamate release. *Hippocampus* **22**, 1350-1362 (2012).

68. Stark, E. et al. Pyramidal Cell-Interneuron Interactions Underlie Hippocampal Ripple Oscillations. *Neuron* **83**, 467-480 (2014).

Suggested Revisions/Comments:

- The introduction section begins with a technology-level approach, but most of the manuscript is centered around a fundamental neuroscience question, making the logic of the manuscript confusing. The manuscript should be re-structured so that the most important aspect of the work is emphasized.*

We thank the reviewer for their comments on the introduction section. Due to the interdisciplinary aspect of this manuscript, we believe that our work has novelty in both the technology development and the neuroscience finding. By developing a novel multi-material thermal tapering method, we have greatly increased the feature density of the probe, along with the scalability. The connectorization is no longer a limiting factor in our probe design. This could open new possibilities of broad distribution of T-DOPe probes in the neuroscience community and a wide selection of new materials, geometrical arrangement, and functionalities that can be embedded in this technology.

Following the reviewer's suggestions, we have revised the introduction to highlight the novelty of this work.

Here, we develop a novel probe technology by merging two cutting edge fabrication approaches, i.e., thermal fiber drawing and multi-material tapering. This scalable probe technology opens new opportunities for broad distribution of flexible opto-electro-pharmacological probes in the neuroscience community. In addition, a wide selection of new materials, geometrical arrangement, and functionalities can be embedded in these tapered fiber probes.

More specifically, the thermal fiber drawing method has emerged as a promising technique for producing scalable multimodality fiber devices at a low cost since 2015.²³⁻²⁹ Such fiber devices are fabricated via a method commonly used in industry to produce optical fibers. The macroscale, multi-material preform is heated until softened, and pulled into hundreds of meters of fibers that can be as thin as a human hair. The fast and simple fabrication process utilizing affordable machinery and soft material results in a cheap, sturdy and biocompatible device. However, an inherent challenge of fiber technology lies in the uniform diameter across the length: there is a tradeoff between minimized sensing tip for biocompatibility and maximized backend tip for easy connection. This has limited the fiber's practicality in neuroscience applications. To overcome this challenge, we developed a novel thermal tapering process (TTP) and a semi-automated connection method, which enable us to fabricate microprobes with high structural and functional complexities and scalability (**fig. 1**, left). Using this approach, the connectorization is no longer a limiting factor in our probe design. It is noteworthy that over the past decades, the tapering of silica has enabled significant advancement in the neuroscience field such as glass micropipettes used in patch clamp recording. And more recently, tapered silica fiber was developed for multisite photometry³⁰ and single neuron recording.⁸ In our work, we demonstrated for the first time, multi-material tapering to create multiplexed, multifunctional neural probes. This novel **Tapered Drug delivery, Optical Stimulation, and Electrophysiology (T-DOpE)** probe allows us to investigate highly complex neural circuitry such as that of the hippocampus in behaving mice performing tasks.

8. LeChasseur, Y. et al. A microprobe for parallel optical and electrical recordings from single neurons in vivo. *Nature Methods* **8**, 319-325 (2011).

23. Canales, A. et al. Multifunctional fibers for simultaneous optical, electrical and chemical interrogation of neural circuits in vivo. *Nature biotechnology* **33**, 277-284 (2015).

24. Loke, G. et al. Digital electronics in fibres enable fabric-based machine-learning inference. *Nature Communications* **12**, 3317 (2021).

25. Marion, J.S. et al. Thermally Drawn Highly Conductive Fibers with Controlled Elasticity. *Advanced Materials* **34**, 2201081 (2022).

26. Jiang, S. et al. Spatially expandable fiber-based probes as a multifunctional deep brain interface. *Nature Communications* **11**, 6115 (2020).

27. Yan, W. et al. Thermally drawn advanced functional fibers: New frontier of flexible electronics. *Materials Today* **35**, 168-194 (2020).

28. Kim, J. et al. Laser Machined Fiber-Based Microprobe: Application in Microscale Electroporation. *Advanced Fiber Materials* **4**, 859-872 (2022).

29. Zhang, Y. et al. Thermally Drawn Stretchable Electrical and Optical Fiber Sensors for Multimodal Extreme Deformation Sensing. *Advanced Optical Materials* **9**, 2001815 (2021).

30. Pisano, F. et al. Depth-resolved fiber photometry with a single tapered optical fiber implant. *Nature Methods* **16**, 1185-1192 (2019).

• *The discussion section is too long. A significant portion of the discussion section should go in the introduction section. Some of the content in the discussion section will help contextualize the motivation of the study if it is moved to the introduction section.*

We thank the reviewer for their perspective on the structure of our manuscript and motivation of our study. Following the reviewer's suggestion, we moved information from the discussion to the introduction to add context to the design of our experiment and the significance of our findings.

We modified the introduction:

The hippocampus plays a major role in memory including for learned spatial locations.^{31, 32} During exploration, hippocampal area CA1 (CA1) pyramidal cell (PYR) activity is organized at the theta timescale (6-11 Hz);^{33, 34} during consummatory behaviors and non-rapid eye movement sleep, large depolarization events drive the generation of sharp wave-ripples (SPW-R, 100-250 Hz). Coordinated activity of PYR organized first by theta followed by SPW-R associated replay is thought to support memory consolidation.³⁵ Further, SPW-Rs are an important electrophysiological marker of learning and memory^{36, 37} and causal roles for SPW-Rs in driving specific behaviors have been demonstrated. Behavioral performance is improved when SPW-Rs are extended in duration;³⁸ while disruption/truncation of SPW-Rs in both wake and non-rem sleep states decreases performance.³⁹⁻⁴² In rodents, cannabinoid type-1 receptor (CB1-R) activation leads to neuronal populations losing their temporally structured co-activity, thought to cause the disruption of hippocampal synchronous activity including epileptic seizures (high frequency oscillations),⁴³⁻⁴⁶ as well as theta oscillations and SPW-Rs.^{47, 48} The cannabinoid induced disruption of theta and SPW-Rs is suggested to be a mechanism behind cannabinoid-associated memory impairment in rodents^{47, 49-52} and humans.⁵³⁻⁵⁵ While a mechanism behind seizure disruption has been identified,⁵⁶ the relationship between the CB1R and memory-supporting rhythms remains unclear. It is believed cannabinoids act through CB1R to impair memory by changing the spiking activity of neurons, either through acting directly on cells expressing CB1R in CA1, or indirectly by acting on presynaptic partners.^{52, 57-62} Thus, the precise mechanisms by which CB1R agonists impair hippocampal rhythms remain unclear.

Theta and SPW-R oscillations require independent mechanisms of generation,^{33-35, 37, 63} and CB1R expression widely varies across cell types and brain areas,^{51, 58-60, 64, 65} further complicating our understanding of specific cellular/synaptic loci at which cannabinoid signaling occurs. Among CA1 neurons, CB1 receptors are expressed on the axon terminals of by CCK+ basket cells, which act to suppress neurotransmitter release,^{58, 59, 66} and on pyramidal cell dendrites, decreasing excitability.⁶⁷ The theta oscillation in CA1 is strongly supported by inputs from the medial septum,^{33, 63} where CB1Rs are widely expressed including on axon terminals projecting to CA1.⁶⁴ Systemic infusion of CB1 agonists disrupts the theta oscillation and organization of spike timing,^{47, 48} but because this method activates CB1Rs throughout the body and brain it is unable to identify mechanistic loci. Thus, it is currently unclear if the activation of CB1Rs within the medial septum, within CA1, or on projections from the medial septum are responsible for theta disruption. The SPW-R oscillation is generated by the reciprocal interaction of excitatory pyramidal cells and local inhibitory interneurons in CA1,⁶⁸⁻⁷¹ initiated by strong excitatory

input from Schaffer collaterals originating in CA3,^{35, 37} whose terminals express CB1 receptors.⁶⁰⁻⁶² SPW-Rs can also be generated experimentally by depolarizing CA1 excitatory neurons.⁶⁸ The cellular-synaptic mechanisms by which cannabinoids disrupt SPW-Rs have remained enigmatic in large part due to the challenging nature of monitoring and manipulating neuronal activity with pharmacological interventions in microcircuits of behaving animals, which our novel probes robustly support. Specifically our devices are able to deliver the synthetic cannabinoid CP-55,940, a validated tool to study the effect of CB1R activation on neuronal activity,^{47-49, 52} to a local circuit which we monitor with electrophysiology and manipulate further with optogenetics.

31. Redish, A.D. Beyond the cognitive map: from place cells to episodic memory. (MIT press, 1999).
32. Buzsáki, G. & Moser, E.I. Memory, navigation and theta rhythm in the hippocampal-entorhinal system. *Nature Neuroscience* **16**, 130-138 (2013).
33. Buzsáki, G. Theta Oscillations in the Hippocampus. *Neuron* **33**, 325-340 (2002).
34. Buzsáki, G. Theta rhythm of navigation: Link between path integration and landmark navigation, episodic and semantic memory. *Hippocampus* **15**, 827-840 (2005).
35. Buzsáki, G. Two-stage model of memory trace formation: A role for noisy brain states. *Neuroscience* **31**, 551-570 (1989).
36. Wilson, M.A. & McNaughton, B.L. Reactivation of Hippocampal Ensemble Memories During Sleep. *Science* **265**, 676-679 (1994).
37. Buzsáki, G. Hippocampal sharp wave-ripple: A cognitive biomarker for episodic memory and planning. *Hippocampus* **25**, 1073-1188 (2015).
38. Fernández-Ruiz, A. et al. Long-duration hippocampal sharp wave ripples improve memory. *Science* **364**, 1082-1086 (2019).
39. Girardeau, G., Benchenane, K., Wiener, S.I., Buzsáki, G. & Zugaro, M.B. Selective suppression of hippocampal ripples impairs spatial memory. *Nature Neuroscience* **12**, 1222-1223 (2009).
40. Gridchyn, I., Schoenenberger, P., O'Neill, J. & Csicsvari, J. Assembly-Specific Disruption of Hippocampal Replay Leads to Selective Memory Deficit. *Neuron* **106**, 291-300.e296 (2020).
41. Ego-Stengel, V. & Wilson, M.A. Disruption of ripple-associated hippocampal activity during rest impairs spatial learning in the rat. *Hippocampus* **20**, 1-10 (2010).
42. Jadhav, S.P., Kemere, C., German, P.W. & Frank, L.M. Awake Hippocampal Sharp-Wave Ripples Support Spatial Memory. *Science* **336**, 1454-1458 (2012).
43. Soltesz, I. et al. Weeding out bad waves: towards selective cannabinoid circuit control in epilepsy. *Nature Reviews Neuroscience* **16**, 264-277 (2015).
44. Cheshier, G.B., Jackson, D.M. & Malor, R.M. Interaction of $\Delta 9$ -tetrahydrocannabinol and cannabidiol with phenobarbitone in protecting mice from electrically induced convulsions. *Journal of Pharmacy and Pharmacology* **27**, 608-609 (1975).
45. Karler, R. & Turkanis, S.A. SUBACUTE CANNABINOID TREATMENT: ANTICONVULSANT ACTIVITY AND WITHDRAWAL EXCITABILITY IN MICE. *British Journal of Pharmacology* **68**, 479-484 (1980).
46. Wallace, M.J., Wiley, J.L., Martin, B.R. & DeLorenzo, R.J. Assessment of the role of CB1 receptors in cannabinoid anticonvulsant effects. *European Journal of Pharmacology* **428**, 51-57 (2001).
47. Robbe, D. et al. Cannabinoids reveal importance of spike timing coordination in hippocampal function. *Nature Neuroscience* **9**, 1526-1533 (2006).
48. Robbe, D. & Buzsáki, G. Alteration of Theta Timescale Dynamics of Hippocampal Place Cells by a Cannabinoid Is Associated with Memory Impairment. *The Journal of Neuroscience* **29**, 12597 (2009).
49. Lichtman, A.H., Dimen, K.R. & Martin, B.R. Systemic or intrahippocampal cannabinoid administration impairs spatial memory in rats. *Psychopharmacology* **119**, 282-290 (1995).
50. Lichtman, A.H. & Martin, B.R. $\Delta 9$ -Tetrahydrocannabinol impairs spatial memory through a cannabinoid receptor mechanism. *Psychopharmacology* **126**, 125-131 (1996).
51. Abush, H. & Akirav, I. Cannabinoids modulate hippocampal memory and plasticity. *Hippocampus* **20**, 1126-1138 (2010).
52. Wise, L.E., Thorpe, A.J. & Lichtman, A.H. Hippocampal CB1 Receptors Mediate the Memory Impairing Effects of $\Delta 9$ -Tetrahydrocannabinol. *Neuropsychopharmacology* **34**, 2072-2080 (2009).
53. Ranganathan, M. & D'Souza, D.C. The acute effects of cannabinoids on memory in humans: a review. *Psychopharmacology* **188**, 425-444 (2006).
54. Ilan, A.B., Smith, M.E. & Gevins, A. Effects of marijuana on neurophysiological signals of working and episodic memory. *Psychopharmacology* **176**, 214-222 (2004).

55. Curran, V.H., Brignell, C., Fletcher, S., Middleton, P. & Henry, J. Cognitive and subjective dose-response effects of acute oral Δ^9 -tetrahydrocannabinol (THC) in infrequent cannabis users. *Psychopharmacology* **164**, 61-70 (2002).
56. Farrell, J.S. et al. In vivo endocannabinoid dynamics at the timescale of physiological and pathological neural activity. *Neuron* **109**, 2398-2403.e2394 (2021).
57. Sullivan, J.M. Mechanisms of Cannabinoid-Receptor-Mediated Inhibition of Synaptic Transmission in Cultured Hippocampal Pyramidal Neurons. *Journal of Neurophysiology* **82**, 1286-1294 (1999).
58. Katona, I. et al. Presynaptically Located CB1 Cannabinoid Receptors Regulate GABA Release from Axon Terminals of Specific Hippocampal Interneurons. *The Journal of Neuroscience* **19**, 4544 (1999).
59. Losonczy, A., Biró, Á.A. & Nusser, Z. Persistently active cannabinoid receptors mute a subpopulation of hippocampal interneurons. *Proceedings of the National Academy of Sciences* **101**, 1362-1367 (2004).
60. Takahashi, K.A. & Castillo, P.E. The CB1 cannabinoid receptor mediates glutamatergic synaptic suppression in the hippocampus. *Neuroscience* **139**, 795-802 (2006).
61. Sandler, R.A., Fetterhoff, D., Hampson, R.E., Deadwyler, S.A. & Marmarelis, V.Z. Cannabinoids disrupt memory encoding by functionally isolating hippocampal CA1 from CA3. *PLOS Computational Biology* **13**, e1005624 (2017).
62. Maier, N. et al. Cannabinoids disrupt hippocampal sharp wave-ripples via inhibition of glutamate release. *Hippocampus* **22**, 1350-1362 (2012).
63. Vandecasteele, M. et al. Optogenetic activation of septal cholinergic neurons suppresses sharp wave ripples and enhances theta oscillations in the hippocampus. *Proceedings of the National Academy of Sciences* **111**, 13535-13540 (2014).
64. Nyíri, G. et al. GABAB and CB1 cannabinoid receptor expression identifies two types of septal cholinergic neurons. *European Journal of Neuroscience* **21**, 3034-3042 (2005).
65. Busquets-Garcia, A., Bains, J. & Marsicano, G. CB1 Receptor Signaling in the Brain: Extracting Specificity from Ubiquity. *Neuropsychopharmacology* **43**, 4-20 (2018).
66. Neu, A., Földy, C. & Soltesz, I. Postsynaptic origin of CB1-dependent tonic inhibition of GABA release at cholecystinin-positive basket cell to pyramidal cell synapses in the CA1 region of the rat hippocampus. *The Journal of Physiology* **578**, 233-247 (2007).
67. Maroso, M. et al. Cannabinoid Control of Learning and Memory through HCN Channels. *Neuron* **89**, 1059-1073 (2016).
68. Stark, E. et al. Pyramidal Cell-Interneuron Interactions Underlie Hippocampal Ripple Oscillations. *Neuron* **83**, 467-480 (2014).
69. Klausberger, T. et al. Brain-state- and cell-type-specific firing of hippocampal interneurons in vivo. *Nature* **421**, 844-848 (2003).
70. Royer, S. et al. Control of timing, rate and bursts of hippocampal place cells by dendritic and somatic inhibition. *Nature Neuroscience* **15**, 769-775 (2012).
71. Dániel, S., Szabolcs, K., Tamás, F.F., Norbert, H. & Attila, I.G. Mechanisms of Sharp Wave Initiation and Ripple Generation. *The Journal of Neuroscience* **34**, 11385 (2014).

REVIEWER COMMENTS

Reviewer #1 (Remarks to the Author):

The authors have addressed all comments from the referees. I feel that the manuscript is suitable for publication in its current form.

Reviewer #2 (Remarks to the Author):

Kim et al describe a novel device that fills an important need in neuroscience for multimodal tools that can simultaneously chemically and optically manipulate the nervous system while recording electrical neural activity. Using this new device, the authors present scientific findings showing that local cannabinoids affect theta and SWR activity in CA1. The authors have addressed most of concerns. A few additional editing points below to strengthen the manuscript.

Major points:

- Characterization of the possible diffusion ranges should be included in the manuscript. The authors include such characterization in the response, but I don't see that it was added to the manuscript: "Depending on the type of chemical, volume, concentration, brain area and duration of the infusion, the likely range is from 0.1-1mm." This is important for potential users to understand the device's spatial resolution and extent capabilities.

- Information about the single unit yield should go in the Results not the Discussion. Specifically, this section:

Though multi-unit activity (MUA) and power in the spike frequency band is observed in all recordings, we used Kilosort and manual curation using Phy to detect sortable units in 13 out of 19 acute sessions (n=15 animals). Detected units ranged from 1-7 units (median = 3) with 57.9% of units identified as putative pyramidal cells, and 42.1% as putative interneurons. Validated spike sorting metrics 74 of Isolation Distance ($\mu=26.9$, $\sigma=9.7$), and percentage of spikes that violate the inter-spike interval ($<2\text{ms}$, $\mu=0.53\%$, $\sigma=0.61\%$) are reported.

- This comment from the authors about blinding and controlling for experimenter bias should be added to the methods: "Due to the nature of the experiment where the researcher had to prepare the drug immediately before the experiment, they were not blinded to the manipulation conditions. However, analysis pipelines including ripple detection and theta power detection were consistent across animals and were not manipulated for experimental condition."

Minor points:

- The introduction is improved but feels like two separate introductions in sequence. The intro would strongly benefit from a transition between the engineering and science sections (eg between 3rd and 4th paragraphs). Even better would be to introduce the scientific question in brief in the first paragraph to motivate the need for this device. This could then be fully elaborated later in the intro (as it currently is). This would help integrate the science and engineering sections, although the authors could achieve this other ways as well.

- I recommend adding this point from the author's response to reviews to the manuscript itself: "A neuroscience lab would likely need to collaborate with an engineering lab to fabricate the customizable device" or similar statement on how a neuroscientist can achieve or access these devices. This statement makes it clear to a neuroscience audience that this is device can be built by an academic lab though special equipment and skills are required

- Are the power spectra in Fig 6c and d from an example animal? The mean of all animals? Please clarify? If it is the mean of all animals, error bars should also be shown.

Reviewer #3 (Remarks to the Author):

The authors did a good job overall in addressing the reviewers' questions. The only thing needs to be further taken care is the response and revision based on the Comment #4 in the first round of revision, quoted as "4. For Figures 5f and 5g where the theta power is plotted against the speed, is the difference between the two linear regressions statistically significant? A two-way ANOVA or a statistics test of the author's choice would be helpful. Also, what is the correlation strength for the linear regressions? Depending on these quantities, the data presented in Figure 5h could be misleading."

It is great that a linear mixed model was incorporated into the analysis. However, there are some things that are still not entirely clear. Is Supplementary Figure 8 supposed to represent multiple mice per plot or are they representative plots? For Supplementary Table 2, are the last 2 columns supposed to have the same headings? In the passage, "The average speed and standard deviation of all 10 sessions suggest that the mice's running behavior was not influenced by drug infusion," is the metric speed or velocity? In Table 2, it says velocity. The interpretation of the data will be different based on whether speed or velocity is used.

In addition, what are Supplementary Figure 8 and Supplementary Table 2 trying to communicate? Is it meant to convey that there is a positive correlation between theta power and running speed and that this is consistent with established results? Is it meant to convey that the running behavior was not influenced by drug infusion by referencing the running velocity? Or is it both? If my interpretation is correct, the p-value of the slopes of the linear mixed model is calculated by testing the null hypothesis that there is zero correlation between the independent and dependent variables. According to the p-values listed in the table, the null hypothesis has not been rejected for any of the mice, and we will need to assume that there is no correlation between theta power and speed. It is mentioned in the manuscript that it is established that there is a positive correlation between theta power and running speed, but this is not reproduced in the manuscript.

If a positive correlation is established between theta wave and running speed, another problem arises. If we assume that theta power is the independent variable and running speed and infusion condition are the dependent variable, confounding variables that can affect the final conclusions are introduced. In other words, how do you know that the infusion type is responsible for the observed differences in theta power? How do you know that running speed or some other variables that affect running speed are not affecting theta power? If we assume that theta power and speed are positively correlated in the data, a simple t-test on one variable (Figure 6e) is not a valid test because we will need to account for interaction effects between any combination of theta power, speed, and infusion type, depending on what is designated as the independent and dependent variables. It may be that, in this field, it is considered standard practice to assume, in such scenarios, that the variables are not confounding. If so, it needs to be clearly stated. It is unlikely that these issues will affect the overall conclusions, but the data needs to be re-analyzed using appropriate statistical tests or approached at a different angle.

Response to reviewers' comments:

We would like to thank all the reviewers for their deep and thorough review of our manuscript, which helped us improve and clarify the manuscript. Reviewers' comments are in *italic*, and the authors' responses are in black. Main text changes are highlighted in red.

Reviewer #1 (Remarks to the Author):

The authors have addressed all comments from the referees. I feel that the manuscript is suitable for publication in its current form.

We appreciate the positive assessment of our manuscript.

Reviewer #2 (Remarks to the Author):

Kim et al describe a novel device that fills an important need in neuroscience for multimodal tools that can simultaneously chemically and optically manipulate the nervous system while recording electrical neural activity. Using this new device, the authors present scientific findings showing that local cannabinoids affect theta and SWR activity in CA1. The authors have addressed most of concerns. A few additional editing points below to strengthen the manuscript.

We thank the reviewer for the positive assessment of our manuscript. As suggested, we have added additional edits to strengthen the manuscript.

Major points:

1. Characterization of the possible diffusion ranges should be included in the manuscript. The authors include such characterization in the response, but I don't see that it was added to the manuscript: "Depending on the type of chemical, volume, concentration, brain area and duration of the infusion, the likely range is from 0.1-1mm." This is important for potential users to understand the device's spatial resolution and extent capabilities.

We thank the reviewer for the helpful suggestion, we have added the following sentence into the discussion section:

Depending on the type of chemical, volume, concentration, brain area and duration of the infusion, the likely range is from 0.1-1mm.

2. Information about the single unit yield should go in the Results not the Discussion. Specifically, this section:

Though multi-unit activity (MUA) and power in the spike frequency band is observed in all recordings, we used Kilosort and manual curation using Phy to detect sortable units in 13 out of 19 acute sessions (n=15 animals). Detected units ranged from 1-7 units (median = 3) with 57.9% of units identified as putative pyramidal cells, and 42.1% as putative interneurons. Validated spike sorting metrics⁷⁴ of Isolation Distance ($\mu=26.9$, $\sigma=9.7$), and percentage of spikes that violate the inter-spike interval (<2ms, $\mu=0.53\%$, $\sigma=0.61\%$) are reported.

We thank the reviewer for the comment and agree this would be more suitable to report in the results section. We moved the specific sentences into results section from the discussion section:

Though multi-unit activity (MUA) and power in the spike frequency band is observed in all recordings, we used Kilosort and manual curation using Phy to detect sortable units in 13 out of 19 acute sessions (n=15 animals). Detected units ranged from 1-7 units (median = 3) with 57.9% of units identified as putative pyramidal cells, and 42.1% as putative interneurons. Validated spike sorting metrics⁷⁴ of Isolation Distance ($\mu=26.9$, $\sigma=9.7$), and percentage of spikes that violate the inter-spike interval (<2ms, $\mu=0.53\%$, $\sigma=0.61\%$) are reported.

3. This comment from the authors about blinding and controlling for experimenter bias should be added to the methods: "Due to the nature of the experiment where the researcher had to prepare the drug immediately before the experiment, they were not blinded to the manipulation conditions. However, analysis pipelines including ripple detection and theta power detection were consistent across animals and were not manipulated for experimental condition."

We thank the reviewer for the comment, and we have included the following sentence in the method section of the manuscript:

Due to the nature of the experiment where the researcher had to prepare the drug immediately before the experiment, they were not blinded to the manipulation conditions. However, analysis pipelines including ripple detection and theta power detection were consistent across animals and were not manipulated for experimental condition.

Minor points:

1. The introduction is improved but feels like two separate introductions in sequence. The intro would strongly benefit from a transition between the engineering and science sections (eg between 3rd and 4th paragraphs). Even better would be to introduce the scientific question in brief in the first paragraph to motivate the need for this device. This could then be fully elaborated later in the intro (as it currently is). This would help integrate the science and engineering sections, although the authors could achieve this other ways as well.

We thank the reviewer for the thoughtful suggestion about the introduction. We have introduced the scientific question and motivation in the first paragraph to motivate the need for this device. We have modified and added these sentences into the introduction:

However, other biological factors such as neurochemistry are intertwined with electrical activity, necessitating simultaneous opto-electro-pharmacological investigations. One such example, is the interaction of exogenous cannabinoids with neural circuitry. With the recent rise in popularity of cannabinoids due to widespread legalization in the United States, this pharmacological-electrophysiological interaction needs to be further investigated. Combining drug infusion with current opto-electric devices has remained challenging. Therefore, developing single devices that can interact with the brain of behaving mammals across such multiple modalities is a critical goal for the field.

2. I recommend adding this point from the author's response to reviews to the manuscript itself: "A neuroscience lab would likely need to collaborate with an engineering lab to fabricate the customizable device" or similar statement on how a neuroscientist can achieve or access these devices. This statement makes it clear to a neuroscience audience that this is device can be built by an academic lab though special equipment and skills are required.

We thank the reviewer for the recommendation, we have added the following sentences to the discussion section:

We are eager to disseminate these devices and will make them available to the neuroscience community upon reasonable request. Additionally, devices can be fabricated by neuroscience labs collaborating with engineering labs with relevant skills and customized equipment (see methods).

3. Are the power spectra in Fig 6c and d from an example animal? The mean of all animals? Please clarify? If it is the mean of all animals, error bars should also be shown.

We appreciate the reviewer's comment on the power spectra. The power spectra from Fig. 6c and 6d are from one session each. The plot in Fig. 6e is the normalized theta power for all the sessions. The theta power was computed by calculating the area under the curve (6-11 Hz).

To clarify this, we have modified the sentences in the results section and figure caption:

The representative power spectrum of baseline (Before infusion; duration: 60 minutes) and after drug infusion (one hour after infusion; duration: 60 minutes) are shown in **fig. 6c**.

Fig. 6d shows an example control session with drug vehicle (VEH), the normalized spectrum of the baseline and after VEH infusion.

Reviewer #3 (Remarks to the Author):

The authors did a good job overall in addressing the reviewers' questions. The only thing needs to be further taken care is the response and revision based on the Comment #4 in the first round of revision, quoted as "4. For Figures 5f and 5g where the theta power is plotted against the speed, is the difference between the two linear regressions statistically significant? A two-way ANOVA or a statistics test of the author's choice would be helpful. Also, what is the correlation strength for the linear regressions? Depending on these quantities, the data presented in Figure 5h could be misleading."

We thank the reviewer's positive comments on the manuscript. We revised the manuscript to further address the comments on the statistical analysis of running behavior (speed) and theta power.

It is great that a linear mixed model was incorporated into the analysis. However, there are some things that are still not entirely clear. Is Supplementary Figure 8 supposed to represent multiple mice per plot or are they representative plots? For Supplementary Table 2, are the last 2 columns supposed to have the same headings? In the passage, "The average speed and standard deviation of all 10 sessions suggest that the mice's running behavior was not influenced by drug infusion," is the metric speed or velocity? In Table 2, it says velocity. The interpretation of the data will be different based on whether speed or velocity is used.

In addition, what are Supplementary Figure 8 and Supplementary Table 2 trying to communicate? Is it meant to convey that there is a positive correlation between theta power and running speed and that this is consistent with established results? Is it meant to convey that the running behavior was not influenced by drug infusion by referencing the running velocity? Or is it both? If my interpretation is correct, the p-value of the slopes of the linear mixed model is calculated by testing the null hypothesis that there is zero correlation between the independent and dependent variables. According to the p-values listed in the table, the null hypothesis has not been rejected for any of the mice, and we will need to assume that there is no correlation between theta power and speed. It is mentioned in the manuscript that it is established that there is a positive correlation between theta power and running speed, but this is not reproduced in the manuscript.

If a positive correlation is established between theta wave and running speed, another problem arises. If we assume that theta power is the independent variable and running speed and infusion condition are the dependent variable, confounding variables that can affect the final conclusions are introduced. In other words, how do you know that the infusion type is responsible for the observed differences in theta power? How do you know that running speed or some other variables that affect running speed are not affecting theta power? If we assume that theta power and speed are positively correlated in the data, a simple t-test on one variable (Figure 6e) is not a valid test because we will need to account for interaction effects between any combination of theta power, speed, and infusion type, depending on what is designated as the independent and dependent variables. It may be that, in this field, it is considered standard practice to assume, in such scenarios, that the variables are not confounding. If so, it needs to be clearly stated. It is

unlikely that these issues will affect the overall conclusions, but the data needs to be re-analyzed using appropriate statistical tests or approached at a different angle.

We thank the reviewer for these insightful comments and we apologize for the confusion. The Supplementary Figure 8 are representative plots (that we decided to remove to avoid confusion). The computed r^2 and p-value of the linear fit and the p-values in the linear mixed model suggest that there is no statistically significant correlation between the running speed and the theta power, which was stated incorrectly in the manuscript (more detailed discussion in the following paragraphs). We also apologize for the typo in the heading of the last two columns which should not be the same. We also split the supplementary table 2 into two separate tables as shown below. The second table is updated and shows the p-values of the linear mixed model. We further quantified the running behavior pre and post infusion in the first table. The Shapiro-Wilk test was conducted to check for normality. Using the paired t-test, we found no significant statistical difference in running behavior before and after the infusion. The mouse is head fixed on a wheel, navigating in 1 dimensional virtual reality. Therefore, in this experimental setup, velocity of the mice is equivalent to speed. We have changed all “velocity” into “speed” in the manuscript to eliminate any possible confusion.

Minmax normalized average speed pre and post infusion (1 hour each)

Infusion type	Mouse ID	Mouse Session	Minmax norm. Avg. speed & std preinfusion	Minmax norm. Avg. speed & std postinfusion	P-value for paired t-test	% time spent running Preinfusion (%)	% time spent running Postinfusion (%)	P-value for paired t-test
Drug Vehicle	m400	m400_221219	0.0473; 0.0825	0.0532; 0.0960	0.9474 (Avg.Speed pre and post infusion is not stastically significant)	8.26	10.78	0.3353 (%time running pre and postinfusion is not stastically significant)
	m401	m401_221224	0.0698; 0.1257	0.0864; 0.1688		13.82	12.57	
	m402	m402_221212	0.0825; 0.1535	0.0661; 0.1485		20.55	13.82	
	m442	m442_231006	0.0500; 0.1199	0.0418; 0.1240		9.76	6.33	
CP-55,940	m402	m402_220926	0.0844; 0.1739	0.0601; 0.1360	0.8293 (Avg.Speed pre and post infusion is not stastically significant)	17.10	12.31	0.8277 (%time running pre and post infusion is not stastically significant)
	m403	m403_221011	0.0251; 0.0918	0.0455; 0.1010		6.89	7.40	
	m404	m404_221018	0.0633; 0.1149	0.0943; 0.1543		11.92	20.11	
	m441	m441_231005	0.0737; 0.1943	0.0723; 0.1736		11.76	12.51	
	m442	m442_231010	0.0658; 0.1603	0.0606; 0.1607		13.30	11.16	

Linear mixed estimate model fitted by using fitme (Matlab)
Power of theta as function of intercept and running speed grouped by infusion conditions

Infusion type	Mouse ID	Mouse Session	P-value for slopes	P-value for intercept
Drug Vehicle	m400	m400_221219	0.56272	8.454e-66
	m401	m401_221224	0.53787	1.4772e-34
	m402	m402_221212	0.19012	1.4671e-105
	m442	m442_231006	0.629	6.1918e-32
CP-55,940	m402	m402_220926	0.37642	3.176e-104
	m403	m403_221011	0.05195	1.5538e-58
	m404	m404_221018	0.69183	4.0685e-109
	m441	m441_231005	0.10348	1.0033e-38
	m442	m442_231010	0.39025	1.6338e-24

It has been well established in our field in freely moving rodents **navigating in the real world** that there is a strong positive correlation between theta power and running speed as well as theta frequency and running speed.(Green & Arduini 1953, Whishaw & Vanderwolf 1973, Vanderwolf 1969, Farland et al., 1975, Gupta et al., 2012, Long et al., 2014, Geisler et al., 2007, Kropff et al., 2021) Robbe et al., found the theta power decreased following systemic administration of cannabinoids, with the confound that systemic cannabinoids are sufficient to reduce motor activity (Long et al., 2010, Bruijnzeel et al., 2016, Kasten et al., 2019, Smoker et al., 2019, Calapai et al., 2022). It is thus possible that the observed reduction in theta power was due to decreased motor activity. To support their claim that effects are due to drug administration, they showed the relationship between theta power and speed is maintained.

However, after reexamination of the linear mixed model, we found no statistically significant relationship between theta power and running speed. This is unsurprising in light of a recent study that showed no statistically significant correlation between theta frequency and running speed in **mice running in virtual reality** (Ravassard et al., 2013). We conducted additional analysis and found that the percentage of time spent running as well as average running speed did not change following intracranial infusion, an important distinction from systemic infusion in previous studies. The relationship between theta power and running speed while interesting, is not a key focus of our study, and thus does not change the interpretation of our results.

We claim that the running behavior is not affected by the local infusion of CP-55,940 and drug vehicle. The linear mixed model shows that the theta power does not have any dependency on the running speed. With these claims, we identify that the decrease in theta power post infusion is due to the effects of CP-55,940.

To ensure clarity regarding this topic, we modified and added the following sentences into the results, discussion, and method sections:

Across all sessions, theta power significantly decreased following CP-55,940 infusion compared to vehicle control (n=5 animals for both VEH, and CP-55,940), with no changes in time spent running or average running speed following infusion (**Supplementary table 2**).

We demonstrated with our T-DOPe probe that local CA1 infusion of CP-55,940 was sufficient to disrupt theta (**fig. 6e**), while time spent running and average running speed did not change (**Supplementary table 2**). We found there was no statistically significant relationship between theta power and running speed in our experiments (**Supplementary table 3**), likely because navigation occurred in a virtual environment.⁸³

The average running speed pre and post infusion was also computed to compare running behavior before and after drug infusion. The percent time spent running was computed for all sessions. The Shapiro-Wilk normality test was adapted to check for normality and the paired t-test was used to observe the running behavior pre and post infusion.

83. Ravassard, P. et al. Multisensory Control of Hippocampal Spatiotemporal Selectivity. *Science* **340**, 1342-1346 (2013).

REVIEWERS' COMMENTS

Reviewer #2 (Remarks to the Author):

The authors have addressed my concerns.

Reviewer #3 (Remarks to the Author):

The revision has fully addressed my comments. I would like to suggest the acceptance of this manuscript for publication as is.